# Spatacsin regulates directionality of lysosome trafficking by promoting the degradation of its partner AP5Z1

**Alexandre Pierga**[1,2,3,4], **Raphaël Matusiak**[1,2,3,4], **Margaux Cauhapé**[1,2,3,4], **Julien Branchu**[1,2,3,4], **Lydia Danglot**[5,6], **Maxime Boutry**[1,2,3,4], **Frédéric Darios**[1,2,3,4]*

1 Sorbonne Université, Paris, France, 2 Paris Brain Institute, ICM, Paris, France, 3 Inserm, U1127, Paris, France, 4 CNRS, UMR 7225, Paris, France, 5 Institute of Psychiatry and Neuroscience of Paris (IPNP), INSERM U1266, Membrane Traffic in Healthy and Diseased Brain, Université Paris Cité, Paris, France, 6 Institute of Psychiatry and Neuroscience of Paris (IPNP), INSERM U1266, Scientific director of NeurImag facility, Université Paris Cité, Paris, France

* frederic.darios@upmc.fr, frederic.darios@icm-institute.org

**Data Availability Statement:** All relevant data are within the paper and its Supporting information files. Two-hybrid data have been deposited to the IntAct database (https://www.ebi.ac.uk/intact/

## Abstract

The endoplasmic reticulum (ER) forms contacts with the lysosomal compartment, regulating lysosome positioning and motility. The movements of lysosomes are controlled by the attachment of molecular motors to their surface. However, the molecular mechanisms by which ER controls lysosome dynamics are still elusive. Here, using mouse brain extracts and mouse embryonic fibroblasts, we demonstrate that spatacsin is an ER-resident protein regulating the formation of tubular lysosomes, which are highly dynamic. Screening for spatacsin partners required for tubular lysosome formation showed spatacsin to act by regulating protein degradation. We demonstrate that spatacsin promotes the degradation of its partner AP5Z1, which regulates the relative amount of spastizin and AP5Z1 at lysosomes. Spastizin and AP5Z1 contribute to regulate tubular lysosome formation, as well as their trafficking by interacting with anterograde and retrograde motor proteins, kinesin KIF13A and dynein/dynactin subunit p150$^{Glued}$, respectively. Ultimately, investigations in polarized mouse cortical neurons in culture demonstrated that spatacsin-regulated degradation of AP5Z1 controls the directionality of lysosomes trafficking. Collectively, our results identify spatacsin as a protein regulating the directionality of lysosome trafficking.

## Introduction

Lysosomes are membrane-limited organelles responsible for the degradation of various cellular substrates. They degrade the content of late endosomes and autophagosomes upon fusion with these subcellular compartments. In addition, they also participate in many other cellular functions, such as cell metabolism and the repair of plasma membranes, as well as adhesion and migration [1]. These diverse functions rely on the cellular localization of lysosomes, as well as their motility and remodeling [2,3]. Accordingly, lysosomes are highly dynamic subcellular compartments [4]. They are retrogradely transported along microtubules upon coupling to cytoplasmic dynein and move anterogradely toward the cell periphery using various

home), accession number IM-29701 The protein interactions from the two hybrid screen using the C-terminal domain of spatacsin (aa1986-2443) as a bait have been submitted to the IMEx (http://www.imexconsortium.org) consortium through IntAct and assigned the identifier IM-29701.

**Funding:** This work was supported by "Investissements d'Avenir" program [ANR-10-IAIHU-06] and [ANR-11-INBS-0011] grants. The NeurImag Imaging Facility team is member of the national infrastructure France-BioImaging supported by the French National Research Agency (ANR-10-INBS-04). The work was supported by funding from the European Research Council (European Research Council Starting [grant No 311149] to F.D.). M.B. received a fellowship from the French Ministry of Research (doctoral school ED3C). A.P. received an ARDOC fellowship from the Région Ile de France (grant 17012953; doctoral school ED3C) and a fellowship from the Fondation pour la Recherche Médicale (grant FDT202001010829). The funders had no role in study design, data collection and analysis, decision to publish, or preparation of the manuscript.

**Competing interests:** The authors have declared that no competing interests exist.

**Abbreviations:** ALR, autophagic lysosome reformation; CHX, cycloheximide; ER, endoplasmic reticulum; MEF, mouse embryonic fibroblast; PFTE, polytetrafluoroethylen; PLA, proximity ligation assay; RT-PCR, reverse transcription PCR; STED, STimulated Emission Depletion.

kinesins [1], changing their cellular distribution. The coordination of this bidirectional transport is particularly important for polarized cells such as neurons. However, the mechanisms regulating the coordination of anterograde or retrograde transports of lysosomes are not clearly established [5].

It has recently emerged that endosomes and lysosomes not only interact with the cytoskeleton but also form functional contacts with other subcellular organelles, in particular, the endoplasmic reticulum (ER). Such contacts with the ER are involved in the filling of lysosomes with $Ca^{2+}$ or the nonvesicular transfer of lipids between the 2 subcellular compartments [6,7]. The interactions of the ER with endosomes and lysosomes also regulate the morphology and trafficking of these subcellular compartments. For example, the interaction of endosomes and lysosomes with the ER controls ER architecture by modulating the formation of the ER network at the cell periphery [8]. Conversely, the ER regulates the distribution of endolysosomes through various mechanisms involving the proteins RNF26, protrudin, and ORP1L [9–11], or it modulates the morphology of endolysosomes by promoting their fission [12,13]. However, the control of endolysosomal dynamics is still only partially understood [14], and the molecular mechanisms regulating lysosome dynamics at the levels of the ER have not been elucidated.

Lysosome function is impaired in various pathological conditions, such as in neurodegenerative diseases [15]. Among them is hereditary spastic paraplegia type SPG11, which is due to loss-of-function mutations in the *SPG11* gene, leading to the absence of spatacsin [16]. The subcellular localization of spatacsin is still debated, as it has been proposed to be localized at the ER, microtubules, or lysosomes [17,18]. However, the loss of spatacsin function has been shown to impair lysosome function and distribution [19–23], suggesting a lysosomal function for this protein. Spatacsin bears a Spatacsin_C domain in its C-terminus, which has been conserved throughout evolution up to plants [24]. However, this domain has no homology in the human genome, suggesting a specific function. Spatacsin interacts with spastizin and AP5Z1, 2 proteins encoded by genes mutated in other forms of hereditary spastic paraplegia, SPG15 and SPG48, respectively [25,26]. Spastizin contains a FYVE domain, which binds to phosphatidylinositol-3-phosphate, allowing its recruitment to lysosomes [27]. AP5Z1 is a subunit of the adaptor protein complex AP5, involved in the sorting of proteins in late endosomes [28]. Loss-of-function mutations in *SPG11*, *SPG15*, or *SPG48* lead to the lysosomal accumulation of material [22,23,29,30]. However, it is not known how the absence of these proteins leads to lysosomal dysfunction and the mechanisms that regulate the interactions between these proteins have not been investigated.

Here, we show that spatacsin is an ER protein that regulates the motility of lysosomes. We show that spatacsin mediates the degradation of AP5Z1, which regulates the relative amounts of AP5Z1 at lysosomes. Moreover, by modulating AP5Z1 levels, spatacsin also indirectly regulates spastizin recruitment at lysosomes. As AP5Z1 and spastizin interact with motor proteins, the dynein/dynactin subunit p150^Glued and the kinesin KIF13A, respectively, spatacsin is likely to regulate lysosome motility. Investigations in polarized axons demonstrate that spatacsin indeed regulates the directionality of lysosomes trafficking.

## Results

### Spatacsin is present in the endoplasmic reticulum and promotes ER–lysosomes contacts

We first investigated the subcellular localization of endogenous spatacsin. Spatacsin has been proposed to contain transmembrane domains that would allow its tight association with membranes [16]. We tested this hypothesis on membrane samples obtained from brains of *Spg11* knockout mice (*Spg11*$^{-/-}$) devoid of spatacsin and compared them to samples of wild-type

mice (*Spg11*<sup>+/+</sup>). We subjected membranes of *Spg11*<sup>+/+</sup> and *Spg11*<sup>−/−</sup> mouse brains to various extraction conditions (i.e., high salt, low pH, high pH, and detergents) [31]. Spatacsin and the integral ER–membrane protein STIM1 were not released from membranes by high salt concentration or low or high pH buffer but were solubilized by the detergent deoxycholate (S1A Fig). Conversely, the membrane-associated protein calreticulin was released from the membranes by high and low pH buffers (S1A Fig). Overall, these data showed that endogenous spatacsin was likely associated to membranes by transmembrane domains.

We then evaluated in which membranes endogenous spatacsin might be present by fractionating mouse brain samples by differential centrifugation (Fig 1A). Spatacsin was present in all but the S3 fraction corresponding to the soluble fraction of cytosolic proteins confirming its anchoring in membranes. Spatacsin was enriched in the P3 microsomal fraction along with the endolysosomal membrane protein Lamp1 and ER proteins STIM1 and calreticulin, whereas the lysosomal peptidase cathepsin D and outer mitochondrial channel VDAC were enriched in the denser P2 fraction and almost absent from the P3 fraction (Fig 1A).

Next, we prepared lysosomes and ER-enriched fractions from *Spg11*<sup>+/+</sup> and *Spg11*<sup>−/−</sup> mouse brains using density gradients. Spatacsin was present in the ER-enriched fraction that contained the ER proteins REEP5 and STIM1 but also contained the lysosomal membrane protein Lamp2 but no cathepsin D (Fig 1B). In contrast, the lysosomal fraction that contained both Lamp2 and cathepsin D was free of ER markers and contained almost no spatacsin, suggesting that spatacsin is not abundant in degradative lysosomes (Fig 1B). Together, these data indicate that endogenous spatacsin is a transmembrane protein present either at the membrane of nondegradative late endosomes or lysosomes, at the ER membrane, or both.

We then explored subcellular localization of spatacsin by immunostaining. Since there are no specific antibodies to spatacsin for immunostaining, we transfected mouse embryonic fibroblasts (MEFs) with a vector allowing the expression of spatacsin with a C-terminal V5 tag or an N-terminal GFP tag. We investigated the localization of spatacsin-V5 or GFP-spatacsin by immunostaining and confocal imaging. Both constructs showed a diffuse distribution of spatacsin that poorly colocalized with late endosomes and lysosomes stained with Lamp1 (Figs 1C and S1B). By contrast, spatacsin-V5 colocalized better with the ER labelled by the expression of GFP-Sec61β. Consistently, STimulated Emission Depletion (STED) imaging showed that spatacsin-V5 was mainly associated with the ER labelled by GFP-Sec61β (Fig 1D, quantification in E) or by REEP5 immunostaining (S1C Fig). Occasionally, spatacsin-V5 colocalized with lysosomes labelled by Lamp1 (Fig 1D and 1E). Of note, lysosomes that were colocalized with spatacsin were in contact with the ER, supporting that spatacsin is localized in the ER.

The absence of spatacsin had no visible impact on the morphology of the ER, as observed by live confocal imaging (S1D Fig). As we have observed that spatacsin is present in the ER and that its loss of function is known to affect lysosomes [20,21], we investigated by transmission electron microscopy if contacts between the ER and lysosomes were altered in absence of spatacsin (Fig 1F). Analysis of MEFs showed that more lysosomes were in contact with the ER in *Spg11*<sup>+/+</sup> than in *Spg11*<sup>−/−</sup> cells (Fig 1F and 1G). Moreover, the proportion of lysosomal membrane that was in contact with the ER was decreased in absence of spatacsin (Fig 1H). The lower proximity between the ER and lysosomes in absence of spatacsin was also detected by confocal microscopy. We transfected *Spg11*<sup>+/+</sup> and *Spg11*<sup>−/−</sup> MEFs with vectors expressing GFP-Sec61β, to label the ER, and Lamp1-mCherry, a marker of late endosomes and lysosomes (henceforth referred to as lysosomes). While lysosomes had the same average size and abundance (S1E and S1F Fig), there was a decrease in the proportion of lysosomes with their area overlapping by more than 30% with the ER staining in absence of spatacsin (Figs 1I, 1J and S1G). Overall, these data show that spatacsin is a protein localized in the ER that promotes contacts between the ER and lysosomes.

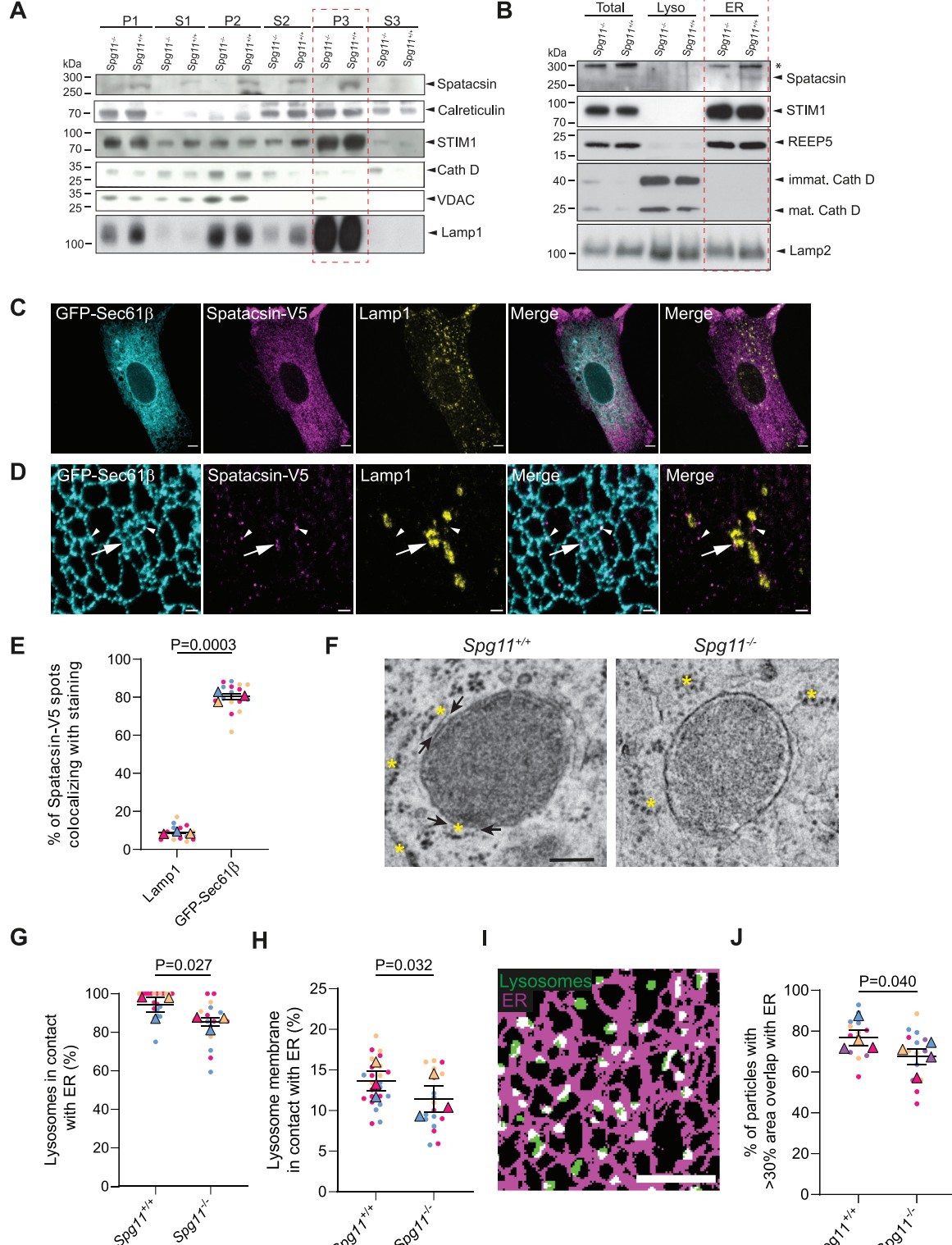

**Fig 1. Spatacsin is present in the endoplasmic reticulum and promotes ER–lysosomes contacts.** (**A**) Western blot analysis of fractions obtained from $Spg11^{+/+}$ and $Spg11^{-/-}$ mouse brains and separated by differential centrifugation. Immunoblots with antibodies raised against spatacsin, the ER proteins STIM1 and calreticulin, the lysosomal proteins cathepsin D (Cath D) and Lamp1, the mitochondrial protein VDAC. Light membrane fractions where spatacsin is enriched are encircled by the red rectangle. (**B**) Western blot analysis of ER- and lysosome-enriched fractions obtained from $Spg11^{+/+}$ and $Spg11^{-/-}$ mouse brains. Immunoblots with antibodies raised against

spatacsin, the ER proteins STIM1 and REEP5, lysosomal hydrolase cathepsin D and lysosomal membrane protein Lamp2. ER fractions are encircled by the red rectangle. The asterisk indicates a nonspecific signal. (**C**) Confocal images of MEFs expressing the ER marker GFP-Sec61β and V5-tagged spatacsin. Cells were immunostained with anti-V5 antibody and the lysosome marker Lamp1. Scale bar: 5 μm. (**D**) STED images of MEFs expressing the ER marker GFP-Sec61β and V5-tagged spatacsin. Cells were immunostained with anti-V5, anti Lamp1, and anti-GFP antibodies. Scale bar: 1 μm. Arrowheads point spatacsin-V5 colocalized with the ER marker GFP-Sec61β. Spatacsin V5 occasionnaly colocalized with the lysosome marker Lamp1 and the ER marker GFP-Sec61β (arrows). (**E**) Quantification of the proportion of V5-tagged spatacsin spots from STED images that are colocalizing either with Lamp1 staining or with GFP-Sec61β expressed in MEFs. Superplot: means and SEM, $N = 18$ cells from 3 independent experiments. Paired $t$ test on the means. (**F**) Transmission electron microscopy images of ER-lysosomes contacts in $Spg11^{+/+}$ and $Spg11^{-/-}$ MEFs. Arrows point toward areas where ER and lysosomes are in contact; asterisks indicate ER. Scale bar: 200 nm. (**G**) Proportion of lysosomes that are less than 30 nm from ER membrane analyzed on transmission electron microscopy images in $Spg11^{+/+}$ and $Spg11^{-/-}$ MEFs. Superplot: means and SEM, $N > 17$ cells from 3 independent experiments. Paired $t$ test on the means. (**H**) Quantification of the proportion of lysosomal membrane that is less than 30 nm from ER membrane analyzed on transmission electron microscopy images in $Spg11^{+/+}$ and $Spg11^{-/-}$ MEFs. Superplot: means and SEM, $N > 17$ cells from 3 independent experiments. * $P = 0.032$, paired $t$ test on the means. (**I**) Binarized image of fibroblasts expressing GFP-Sec61β and Lamp1-mCherry. Note the overlap (white) between the ER (magenta) and lysosome (green) masks. Scale bar: 5 μm. (**J**) Quantification of the proportion of lysosomes that have an area overlapping with the ER > 30% in $Spg11^{+/+}$ and $Spg11^{-/-}$ MEFs. Superplot: means and SEM, $N = 13$ cells from 4 independent experiments. Paired $t$ test on the means. The raw data underlying panels E, G, H, and J can be found in S1 Data file.

## Spatacsin regulates the dynamics of lysosomes that are moving along the ER

Since spatacsin promotes the formation of contacts between the ER and lysosomes that are important to modulate lysosome function [4,14], we evaluated how the loss of spatacsin may affect the lysosomes and their interaction with the ER.

Using MEFs transfected with a vector expressing Lamp1-mCherry, we observed by live imaging a higher number of lysosomes with tubular shape in $Spg11^{+/+}$ compared to $Spg11^{-/-}$ cells (Fig 2A and 2B), suggesting that spatacsin is required for the formation of tubular lysosomes. We also observed a higher number of tubular lysosomes in $Spg11^{+/+}$ than in $Spg11^{-/-}$ MEFs when they were labelled with the acidic marker Lysotracker, with a pulse of Dextran-Texas Red followed by a long chase, or with DQ-BSA, which fluoresces upon degradation by lysosomal hydrolases [32] (S2A and S2B Fig). These results suggests that tubular lysosomes that are fewer in absence of spatacsin represent a population of catalytically active lysosomes.

To further investigate the properties of tubular lysosomes, we analyzed live images of MEFs expressing Lamp1-mCherry and GFP-Sec61β. This showed that the overlap of lysosome and ER staining was greater for tubular lysosomes than round lysosomes, suggesting that tubular lysosomes were in closer proximity to the ER (Fig 2C). Accordingly, STED microscopy showed that the tubular lysosomes were entangled in a network of ER tubules (Fig 2D). Live imaging also showed that tubular lysosomes moved along the ER network (Fig 2E and S1 Video), suggesting that tubular lysosomes are closely associated with the ER.

Since spatacsin promotes contacts between ER and lysosomes and contributes to the formation of tubular lysosomes, we evaluated the properties of tubular lysosomes. Tracking individual lysosomes in wild-type and $Spg11^{-/-}$ MEFs by live imaging showed tubular lysosomes to move, on average, faster than round lysosomes (Fig 2F). The proportion of lysosomes with a speed >0.3 μm/s, corresponding to microtubule-dependent movement for these organelles [33], was higher among the tubular than round lysosomes (S2C Fig), suggesting that tubular lysosomes are highly motile and dynamic. Comparison of the speed of lysosomes in $Spg11^{+/+}$ and $Spg11^{-/-}$ MEFs showed that tubular lysosomes moved faster in $Spg11^{+/+}$ than in $Spg11^{-/-}$ MEFs (Fig 2G and S2 and S3 Videos), whereas the dynamics of round lysosomes was not affected (S2D Fig). Finally, tracking of individual lysosomes showed that lysosomes can change morphology from round to tubular and from tubular to round. The transition between the 2 states was associated with a strong change in the speed of displacement that was greater in

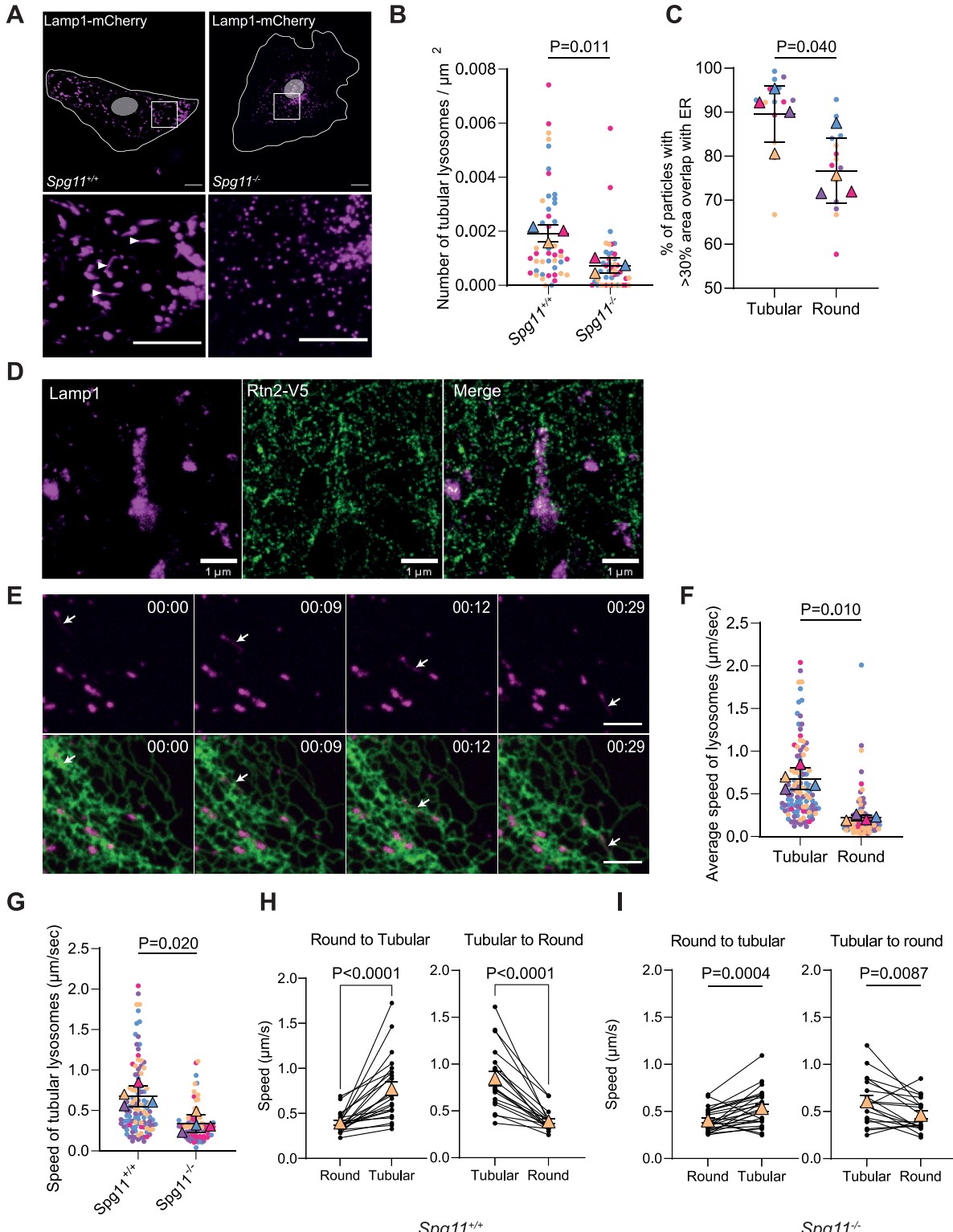

**Fig 2. Spatacsin regulates the dynamics of tubular lysosomes that are moving along the ER.** (**A**) Lamp1-mCherry expression in $Spg11^{+/+}$ and $Spg11^{-/-}$ MEFs imaged by spinning disk confocal microscopy. Note the presence of tubular lysosomes in $Spg11^{+/+}$ MEFs (white arrowheads). Scale bar: 5 μm. (**B**) Quantification of the number of tubular lysosomes in $Spg11^{+/+}$ and $Spg11^{-/-}$ MEFs. Superplot: means and SEM, $N = 45$ cells from 3 independent experiments. Paired $t$ test on the means. (**C**) Quantification of the proportion of lysosomes that have an area overlapping with the ER > 30% based on their shape. Superplot: means and SEM, $N = 14$ cells from 4 independent experiments.

Paired $t$ test on the means. (**D**) STED image of a tubular lysosome stained with Lamp1 antibody and its close interaction with the ER tubular network stained by anti-V5 antibody targeting Reticulon2-V5 (Rtn2-V5) expressed in wild-type MEFs. Scale bar: 1 μm. (**E**) Snapshot images of live imaging of a wild-type MEF transfected with Lamp1-mCherry (Magenta) and GFP-Sec-61β (Green). A tubular lysosome trafficking along the ER tubule network is indicated by an arrow. Scale bar: 5 μm. Time is expressed in min:s. (**F**) Average speed of tubular and round lysosomes in wild-type MEFs. Superplot: means and SEM, $N > 97$ cells from 4 independent experiments. Paired $t$ test on the means. (**G**) Quantification of the average speed of tubular lysosomes in $Spg11^{+/+}$ and $Spg11^{-/-}$ MEFs. Superplot: means and SEM, $N > 77$ cells from 4 independent experiments. Paired $t$ test on the means. (**H**) Quantification of the instant speed of lysosomes in $Spg11^{+/+}$ during their transition from round to tubular or tubular to round shape. Mean and SEM, $N = 23$ pairs of lysosomes from 12 independent MEFs. Paired $t$ tests. (**I**) Quantification of the instant speed of lysosomes in $Spg11^{-/-}$ during their transition from round to tubular or tubular to round shape. Mean and SEM, $N = 19$ pairs of lysosomes from 5 independent MEFs. Paired $t$ tests. The raw data underlying panels B, C, F, G, H, and I can be found in S1 Data file.

$Spg11^{+/+}$ than in $Spg11^{-/-}$ MEFs (Fig 2H and 2I) This showed that tubular lysosomes were more motile than round lysosomes, but their average speed was lower in absence of spatacsin.

Overall, these data suggest that tubular lysosomes represent a population of catalytically active and highly dynamic lysosomes moving in close contact with the ER. As spatacsin plays a key role in the formation and motility of tubular lysosomes, we next aimed to investigate how it could regulate their dynamics.

## Spatacsin regulates the function of lysosomes by interacting with proteins involved in degradation pathways

We used MEFs derived from a mouse line in which exons 32 to 34 of $Spg11$ are spliced out (S3A–S3C Fig) to elucidate the molecular mechanisms by which spatacsin controls these lysosomal phenotypes. Such splicing retained the reading frame and led to the expression of a protein called spatacsin$^{Δ32-34}$, which lacks a domain of 170 amino acids, partially deleting the conserved Spatacsin_C domain (Fig 3A). Western blot analysis of brains obtained from $Spg11^{+/+}$, Spg$11^{-/-}$, and $Spg11^{Δ32-34/Δ32-34}$ mice showed the latter strain to express a spatacsin protein that is smaller than the wild-type protein (S3D Fig).

To evaluate the importance of the domain deleted in $Spg11^{Δ32-34/Δ32-34}$ mice, we performed behavioral and immunohistochemical evaluations of these animals and compared them to $Spg11^{-/-}$ mice that present cognitive impairment and motor dysfunction [22]. To evaluate motor dysfunction, $Spg11^{Δ32-34/Δ32-34}$ mice were challenged by an accelerating rotarod. They remained on the apparatus for significantly less time than wild-type mice but remained on the apparatus longer than Spg$11^{-/-}$ mice (S3E Fig). The alternation in the Y-maze test was used to monitor cognitive function [22]. $Spg11^{+/+}$ mice explored a new environment (alternation) in 75% of the trials. In contrast, $Spg11^{Δ32-34/Δ32-34}$ and $Spg11^{-/-}$ mice showed a lower alternation than the $Spg11^{+/+}$ mice from the age of 4 months (S3F Fig). Together, these data demonstrate that the domain encoded by exons 32 to 34 of $Spg11$ plays a critical role as expression of a protein lacking this domain is sufficient to impair the cognitive and motor functions. We then investigated whether the absence of the domain encoded by exons 32 to 34 contributed to accumulation of autophagic material, as observed in $Spg11$ knockout mice [22,23]. We observed accumulation of the autophagic marker p62 in the motor cortex in 8-month-old $Spg11^{-/-}$ and $Spg11^{Δ32-34/Δ32-34}$ mice, whereas $Spg11^{+/+}$ mice presented almost no p62 accumulation (S3G Fig). Together, these data demonstrated that the domain encoded by exons 32 to 34 are required for the proper function of spatacsin. We therefore investigated the role of spatacsin$^{Δ32-34}$ on lysosomes.

Overexpression of spatacsin$^{Δ32-34}$ with a C-terminal V5 tag in MEFs showed diffuse and ER-associated localization like that of full-length spatacsin (S3H Fig). Like $Spg11^{-/-}$ MEFs, $Spg11^{Δ32-34/Δ32-34}$ fibroblasts had fewer tubular lysosomes than wild-type cells and had altered

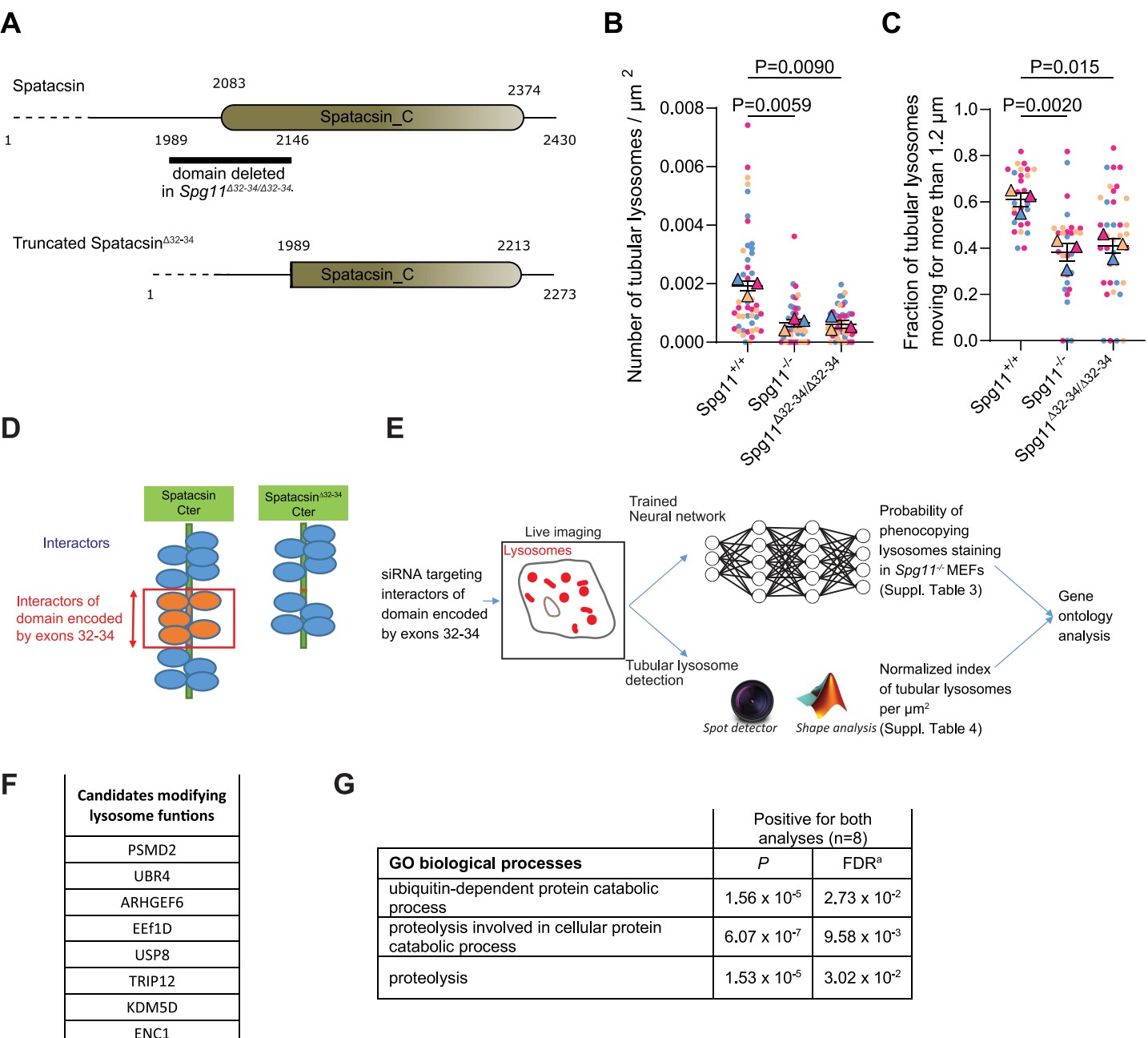

**Fig 3. Spatacsin regulates lysosome function by interacting with partners involved in protein degradation.** (**A**) Representation of the Spatacsin_C domain and the region truncated in the $Spg11^{\Delta32-34/\Delta32-34}$ mouse model. (**B**) Quantification of the number tubular lysosomes in MEFs devoid of spatacsin or expressing truncated spatacsin. Superplot: means and SEM, $N > 44$ cells from 3 independent experiments. RM one-way ANOVA on the means followed by Dunnett's multiple comparisons test. (**C**) Quantification of the proportion of tubular lysosomes moving > 1.2 μm over 1 minute in $Spg11^{+/+}$, $Spg11^{-/-}$, and $Spg11^{\Delta32-34/\Delta32-34}$ MEFs. Superplot: means and SEM, $N > 28$ cells from 3 independent experiments. RM one-way ANOVA on the means followed by Dunnett's multiple comparisons test. (**D**) Scheme showing the strategy to identify interactors of the domain of spatacsin encoded by exons 32 to 34 of *SPG11*. Yeast two-hybrid screens were performed with the C-terminal domain of human spatacsin (aa 1,943 to 2,443) or the same domain missing exons 32 to 34 as bait. Interactors specifically interacting with the spatacsin domain encoded by exons 32 to 34 (orange) were selected. (**E**) Design of the screening process for interactors of the spatacsin domain encoded by exons 32 to 34 of *SPG11*. Each interactor was down-regulated by siRNA in wild-type MEFs, and lysosomes were imaged by spinning disk confocal microscopy. The effect of the siRNAs was analyzed using an unbiased method (trained neural network) or directed analysis to quantify the presence of tubular lysosomes. (**F**) Table of the spatacsin interactors identified by the trained neural network and the directed analysis as modifiers of lysosomal function. (**G**) Table summarizing the pathways identified by gene ontology analysis as being significantly enriched in the list of genes identified in common by neural network analysis and directed analysis. FDR, false discovery rate. The raw data underlying panels B and C can be found in S1 Data file.

tubular lysosomes dynamics (Fig 3B and 3C). The difference in the dynamics of tubular lysosomes was validated by automated tracking, which showed tubular lysosomes to travel longer distance during 1 minute in $Spg11^{+/+}$ than in $Spg11^{-/-}$ and $Spg11^{\Delta32-34/\Delta32-34}$ MEFs (Fig 3C). We used this method to analyze lysosomal dynamics in subsequent experiments. Overall, these results suggest that the spatacsin domain encoded by exons 32 to 34 of $Spg11$ plays an important role in the formation and dynamics of tubular lysosomes.

We next aimed to define the molecular action of spatacsin in the formation of tubular lysosomes. We thus sought to identify proteins that bind to the domain encoded by exons 32 to 34 of $Spg11$. We performed a two-hybrid screen with the C-terminal region of human spatacsin (aa 1,943 to 2,443, containing the Spatacsin_C domain) and a second screen with the same C-terminal fragment in which the amino acids encoded by exons 32 to 34 were deleted (S1 and S2 Tables). Comparison of the 2 screens identified several proteins that potentially bind directly to the domain encoded by exons 32 to 34 (Fig 3D).

Among the proteins that could potentially bind to the domain encoded by exons 32 to 34, we aimed to identify those important for the regulation of lysosome function. Thus, we down-regulated each identified partner in wild-type MEFs using siRNA and analyzed the consequences on lysosomes, which were imaged by spinning disk confocal microscopy after staining with Texas Red–conjugated Dextran.

We used 2 methods to quantify the effect of siRNA on lysosomes (Fig 3E). First, we developed an unbiased classification method to discriminate between lysosomal staining in $Spg11^{+/+}$ and $Spg11^{-/-}$ MEFs, which are expected to represent extreme phenotypes linked to function of spatacsin. This was performed using large series of images to train a neural network that exploited all parameters of the lysosomal staining in images in $Spg11^{+/+}$ and $Spg11^{-/-}$ MEFs. The neural network classification was validated on an independent set of images of lysosomes obtained from $Spg11^{+/+}$ and $Spg11^{-/-}$ MEFs. The trained neural network was then used to predict the probability of the cell to be considered as a $Spg11^{-/-}$ fibroblast for each image of fibroblasts transfected with siRNA. This prediction was based on the parameters the network identified during its training phase as significant to discriminate between $Spg11^{+/+}$ and $Spg11^{-/-}$ lysosomal stainings. In parallel, we performed a directed analysis that automatically detected tubular lysosomes. For both methods, we evaluated how well the down-regulation of each candidate using siRNA in wild-type MEFs phenocopied the lysosomal phenotype of $Spg11^{-/-}$ MEFs. We compared the effect of each siRNA with that of 3 independent siRNAs down-regulating $Spg11$ (S4A and S4B Fig). The neural network approach identified 28 genes and directed analysis identified 11 genes that, upon down-regulation by siRNA, were at least as effective as $Spg11$ siRNA to phenocopy $Spg11^{-/-}$ MEFs (Fig 3E and S3 and S4 Tables). Eight genes were identified in common by both analyses (Figs 3F, S4C and S4D), indicating that the products of these genes may interact with spatacsin to regulate its function in lysosomes.

Gene ontology analysis of the list of identified candidates pointed toward a role of the ubiquitin-dependent protein catabolic process and proteolysis in modulation of the lysosomal phenotype (Fig 3G), suggesting that the action of spatacsin on lysosomes may be linked to protein degradation pathways.

## Spatacsin promotes degradation of AP5Z1 by lysosomes

Our screening approach led us to investigate how the control of tubular lysosome formation and dynamics by spatacsin relied on a degradation pathway. However, exploration of the interactome of the spatacsin domain encoded by exons 32 to 34 revealed no binding partners with endolysosomal localization. We therefore hypothesized that the degradation-dependent

regulation may act on proteins present in the endolysosomal system. Importantly, 2 partners of spatacsin, spastizin and AP5Z1, colocalize with lysosomes [18].

We first investigated whether spastizin or AP5Z1 might be degraded in a spatacsin-dependent manner. We observed that overexpression of spatacsin-GFP in wild-type MEFs lowered levels of AP5Z1, while levels of spastizin were unaffected (Fig 4A). Of note, overexpression of spatacsin-GFP lowered the levels of another subunit of the AP5 complex, AP5M1 (Figs 4A and S5A). Overexpression of spatacsin strongly increased the amount of ubiquitinated AP5Z1 (Fig 4B), which may contribute to its degradation. Interestingly, overexpressed spatacsin itself was also ubiquitinated (Fig 4B), suggesting its levels may also be regulated in an ubiquitin-dependent manner. To test whether endogenous spatacsin may promote the degradation of AP5Z1 or spastizin, we overexpressed AP5Z1-His and Spastizin-HA in $Spg11^{+/+}$ and $Spg11^{-/-}$ MEFs. We observed higher expression of AP5Z1-His in $Spg11^{-/-}$ than in $Spg11^{+/+}$ MEFs, whereas there was no significant difference in the levels of spastizin-HA (S5B Fig), suggesting that endogenous spatacsin promotes degradation of AP5Z1.

The decrease in AP5Z1 levels upon spatacsin-GFP overexpression was blocked when MEFs were treated with the inhibitor of lysosomal acidification, bafilomycin, but not with the proteasome inhibitor MG132 (Fig 4C), suggesting that AP5Z1 was degraded by lysosomes in a spatacsin-dependent manner. To test this hypothesis, we expressed AP5Z1 fused to both GFP and mCherry and analyzed by live imaging the colocalization of fluorescence with the acidic lysosome marker, lysotracker (Figs 4D and S5C). In wild-type MEFs, mCherry was mainly colocalized with lysosomes, and a significant amount of signal was also present in the cytosol. In contrast, GFP that is sensitive to pH was poorly colocalized with lysosomes, indicating that GFP-mCherry-AP5Z1 colocalized with lysosome was mainly inside the acidic subcellular compartment (Fig 4D). Most GFP signal was in cytosol, suggesting that AP5Z1 is a cytosolic protein that transiently associates with lysosomes. However, GFP was slightly enriched around lysosomes (inset Fig 4D), and some AP5Z1 may thus be localized at the lysosomal surface. In $Spg11^{-/-}$ MEFs, the amount of GFP colocalized with lysosomes was higher than in wild-type MEFs, suggesting that AP5Z1 accumulated at the cytosolic surface of lysosomes in absence of spatacsin (Fig 4D and 4E). Of note, GFP signal was recovered and fully colocalized by mCherry signal when MEFs were treated with bafilomycin to neutralize lysosomal pH, confirming that AP5Z1 was internalized into acidic lysosomes (S5C Fig).

Together, these results indicated that spatacsin contributes the degradation of AP5Z1 by promoting its translocation inside lysosomes.

## UBR4 contributes to spatacsin stability and AP5Z1 degradation

We then tested whether the domain encoded by exons 32 to 34 is important for the degradation of AP5Z1, by overexpressing spatacsin$^{\Delta 32-34}$-GFP. The lower levels of AP5Z1 were not observed upon overexpression of spatacsin$^{\Delta 32-34}$-GFP, suggesting that the domain encoded by exons 32 to 34 is important to promote the degradation of AP5Z1 (Fig 4A). The domain encoded by exons 32 to 34 notably interacts with UBR4, which was identified in our screen as a mediator of lysosomal function (Fig 3F). We confirmed by co-immunoprecipitation that UBR4 interacts with the C-terminal domain of spatacsin, but not the domain lacking the fragment encoded by exons 32 to 34 of $Spg11$ (Fig 5A). Down-regulation of UBR4 partially prevented the degradation of AP5Z1 mediated by spatacsin-GFP overexpression (Fig 5B) and also reduced the levels of overexpressed spatacsin-GFP (Fig 5B), suggesting that UBR4 may modulate the levels of both proteins. Of note, down-regulation of UBR4 also led to higher levels of AP5Z1 in control MEFs (Fig 5B), suggesting that UBR4 may regulate degradation of AP5Z1 by endogenous spatacsin. To further address the role of UBR4, we down-regulated UBR4 and

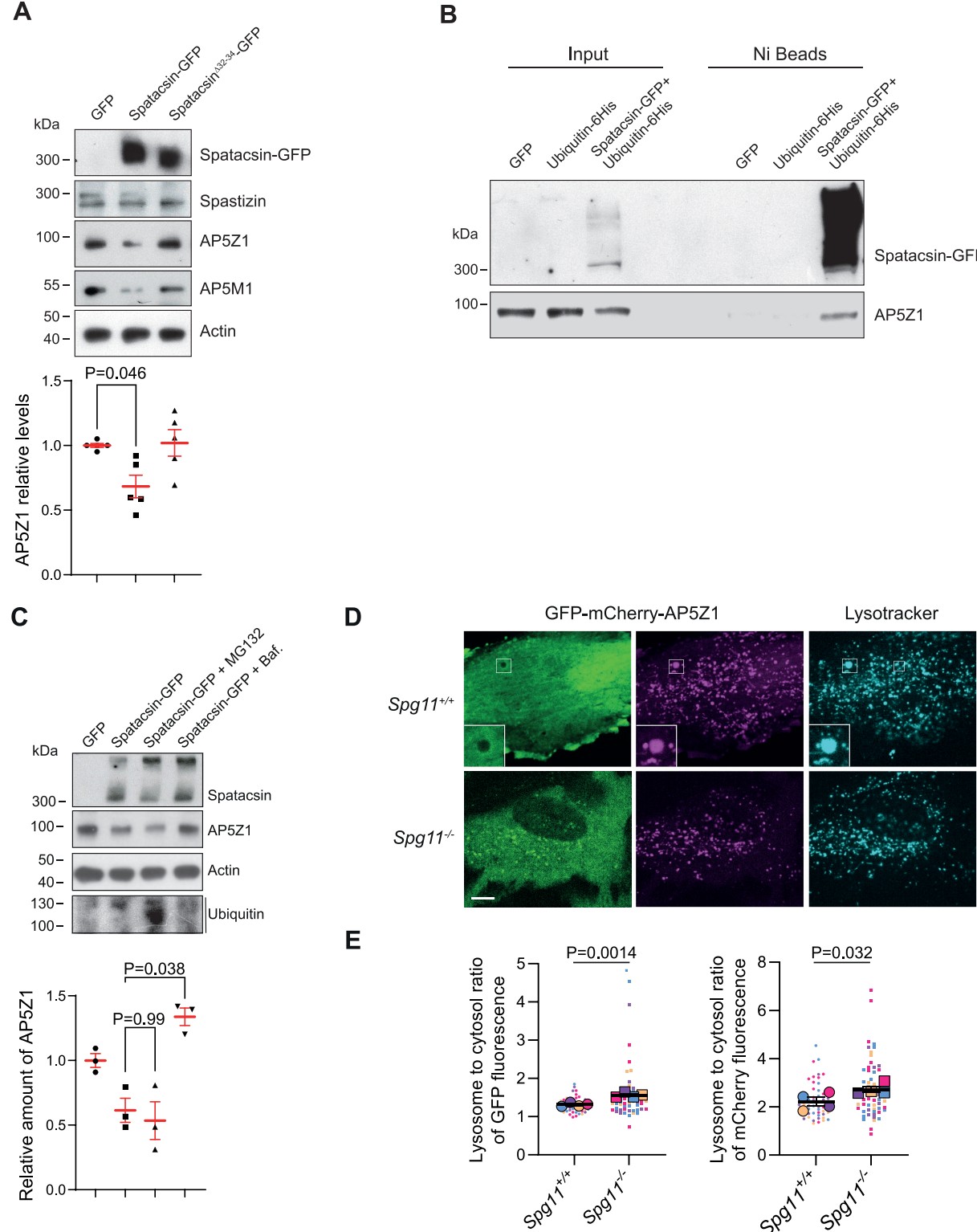

**Fig 4. Spatacsin promotes degradation of AP5Z1 by lysosomes.** (**A**) Western blots showing levels of spatacsin-GFP, spastizin, AP5Z1, AP5M1, and actin in MEFs overexpressing GFP, spatacsin-GFP, or spatacsin$^{\Delta 32-34}$-GFP. Quantification of AP5Z1 levels by western blot: means and SEM, $N$ = 5 independent experiments. Kruskal–Wallis test. (**B**) Western blots showing levels of AP5Z1 and spatacsin-GFP in inputs and in Ni-NTA beads pulldown performed in MEFs transfected with vectors expressing GFP, 6-His-Ubiquitin or 6-His-Ubiquitin together with spatacsin-GFP. Pulldown of 6-His-tagged proteins with Ni-NTA beads shows polyubiquitination of spatacsin-GFP, as well as modification of

AP5Z1 with 6-His Ubiquitin when spatacsin-GFP is overexpressed. Note that ubiquitinated AP5Z1 appears with a slightly lower apparent molecular weight. (**C**) Western blot monitoring expression levels of AP5Z1 in wild-type MEFs overexpressing spatacsin-GFP and treated for 16 hours with either MG132 (15 μM) or bafilomycin (Baf., 100 nM). Actin immunoblot was used as loading control. Ubiquitin immunoblot showed accumulation of ubiquitylated proteins upon MG132 treatment. Quantification of AP5Z1 levels by western blot: means and SEM, $N = 3$ independent experiments. Kruskal–Wallis test. (**D**) Live images of $Spg11^{+/+}$ and $Spg11^{-/-}$ MEFs expressing GFP-mCherry-AP5Z1 and stained with Lysotracker Blue. Inset shows the mCherry fluorescence colocalized with acidic lysosomes labelled by lysotracker while the GFP signal is located on the surface of lysosomes. Scale bar: 10 μm. (**E**) Quantification of the ratio of GFP and mCherry fluorescence in lysosomes compared to fluorescence in the cytosol in $Spg11^{+/+}$ and $Spg11^{-/-}$ MEFs transfected with a vector expressing GFP-mCherry-AP5Z1. Superplot: means and SEM, $N > 50$ cells in 4 independent experiments. Paired $t$ test on the means. The raw data underlying panels A, C, E, and I can be found in S1 Data file.

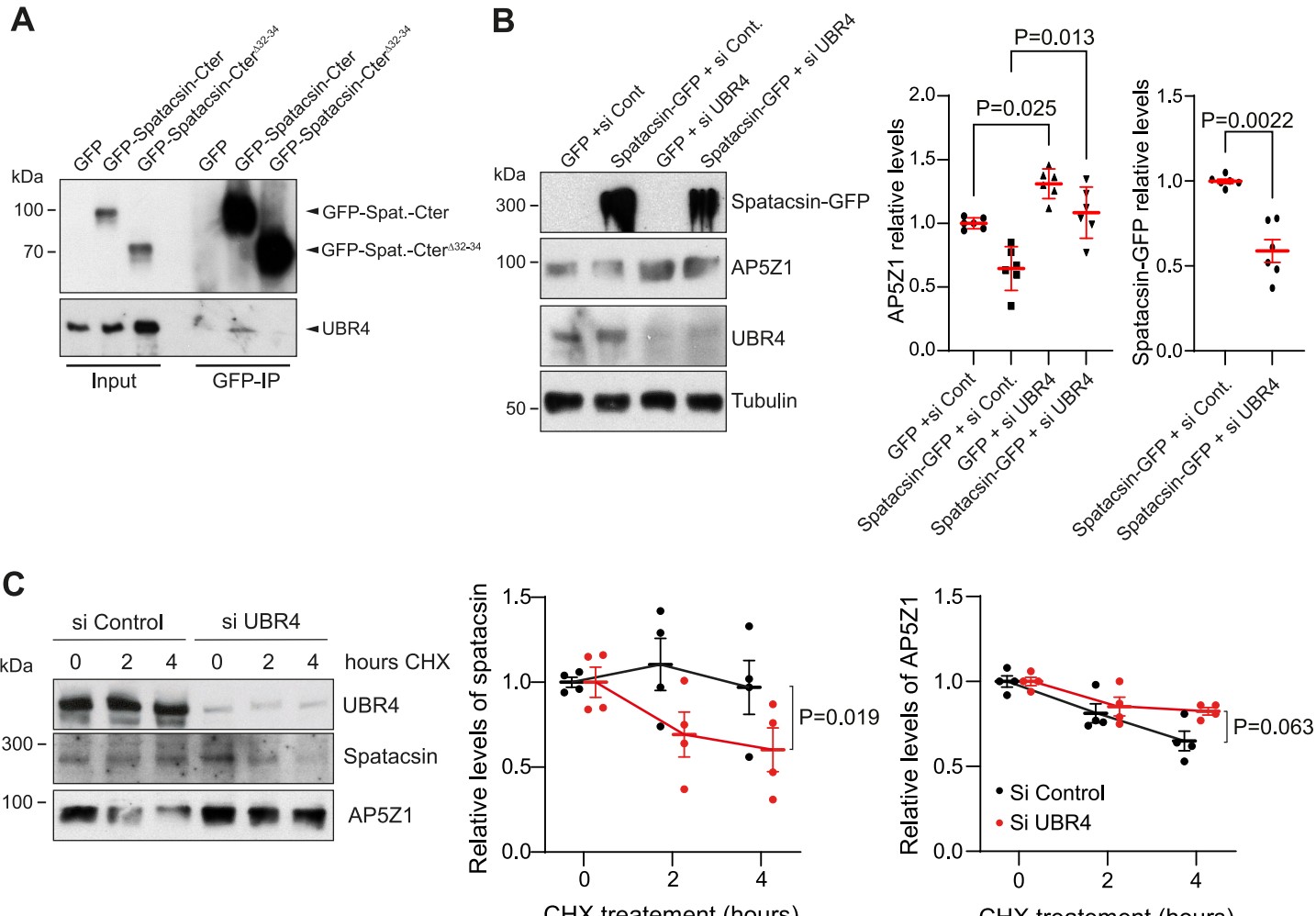

**Fig 5. UBR4 contributes to spatacsin stability and AP5Z1 degradation.** (**A**) Western blots showing co-immunoprecipitation of UBR4 with the C-terminal domain of spatacsin (aa 1,943–2,443, GFP-spatacsin-Cter) but not the construct lacking amino acids encoded by exons 32 to 34 (GFP-spatacsin-Cter$^{\Delta32-34}$). Input represents 5% of lysate added to the immunoprecipitation assay. (**B**) Western blot showing levels of AP5Z1 in wild-type MEFs transfected with vectors expressing GFP, or spatacsin-GFP with control siRNA or siRNA down-regulating UBR4. Means and SEM, $N = 6$ independent experiments. Kruskal–Wallis test (multiple comparison) and Mann–Whitney (comparison of 2 groups). (**C**) Western blots showing levels of AP5Z1 and spatactin after inhibiting protein translation with 10 μg/ml cycloheximide (CHX) in MEFs transfected with control siRNA or siRNA down-regulating UBR4. Means and SEM, $N = 4$ independent experiments. Two way-ANOVA. The raw data underlying panels B, C, and I can be found in S1 Data file.

monitored the levels of endogenous AP5Z1 and spatacsin after inhibition of protein translation with cyloheximide. Levels of endogenous AP5Z1 decreased slightly slower upon UBR4 down-regulation (Fig 5C), consistent with a role of UBR4 to control degradation of AP5Z1. Interestingly, down-regulation of UBR4 enhanced the decrease in endogenous spatacsin levels (Fig 5C), suggesting that the presence of UBR4 is important for spatacsin stability.

Together, these data suggested that UBR4 is important for spatacsin stability, and its down-regulation likely prevented the degradation of AP5Z1 mediated by spatacsin, leading to change in relative levels of both proteins.

## Spatacsin promotes spastizin recruitment to lysosomes

We then investigated whether spatacsin may also regulate the function of its other interacting partner spastizin. We observed weaker colocalization of spastizin-GFP with Lamp1-mCherry in *Spg11*$^{-/-}$ than *Spg11*$^{+/+}$ MEFs by live imaging (Fig 6A and 6B), consistent with previous observation [27]. To test whether spastizin interaction with spastizin was required for spastizin localization to lysosomes, we performed a proximity ligation assay in MEFs transfected with spatacsin-V5 and spastizin-HA. This assay showed that the sites where spatacsin interacted with spastizin colocalized with the ER labeled by GFP-Sec61β and lysosomes labelled by Lamp1 immunostaining (Figs 6C and S6A). This suggests that the spatacsin–spastizin interaction occurs at contact sites between the ER and lysosomes to allow spastizin recruitment to lysosomes.

We then investigated whether the degradation of AP5Z1 may have an impact on the interaction of spatacsin with spastizin by co-immunoprecipitation. To mimic the consequence of its degradation, we down-regulated AP5Z1 using siRNA (Fig 6D). Down-regulation of AP5Z1 favored the co-immunoprecipitation of spatacsin with spastizin (Fig 6D). Conversely, overexpression of AP5Z1 lowered the amount of spastizin co-immunoprecipitated with spatacsin (S6B Fig), suggesting that interaction of spatacsin with spastizin relied on the relative levels of AP5Z1 and that AP5Z1 may compete with spastizin to bind spatacsin. Furthermore, down-regulation of AP5Z1 promoted spastizin colocalization with lysosomes (Fig 6E), whereas overexpression of AP5Z1 partially prevented association of spastizin with lysosomes (S6C Fig).

Overall, these data suggested that AP5Z1 competes with spastizin to interact with spatacsin and that excess of AP5Z1 prevents association of spastizin with lysosomes that is mediated by spatacsin.

## Spastizin and AP5Z1 are required for the formation of tubular lysosomes and interact with motor proteins

We then investigated whether AP5Z1 and spastizin regulated the formation and dynamics of tubular lysosomes. To evaluate whether AP5Z1 levels impacted lysosomes dynamics, we cotransfected wild-type MEFs with Lamp1-mCherry and either a vector expressing GFP-AP5Z1 or siRNA down-regulating AP5Z1 (Figs 7A and S7A–S7C). Both overexpression and down-regulation of AP5Z1 decreased the number of tubular lysosomes (Figs 7A and S7B) and impaired their dynamics in wild-type MEFs (Figs 7C and S7C). This observation suggested that AP5Z1 was required to regulate tubular lysosome motility, but its level must be tightly regulated.

To evaluate the role of spastizin at the lysosomes, we down-regulated spastizin using siRNA (S7D Fig). This condition decreased the number of tubular lysosomes and their dynamics (Fig 7B and 7C). Similar results were obtained when we treated wild-type MEFs with the PI3 kinase inhibitor wortmannin that prevented the lysosomal localization of spastizin (S7E and S7F Fig). Therefore, both AP5Z1 and spastizin contributed to regulate tubular lysosome motility.

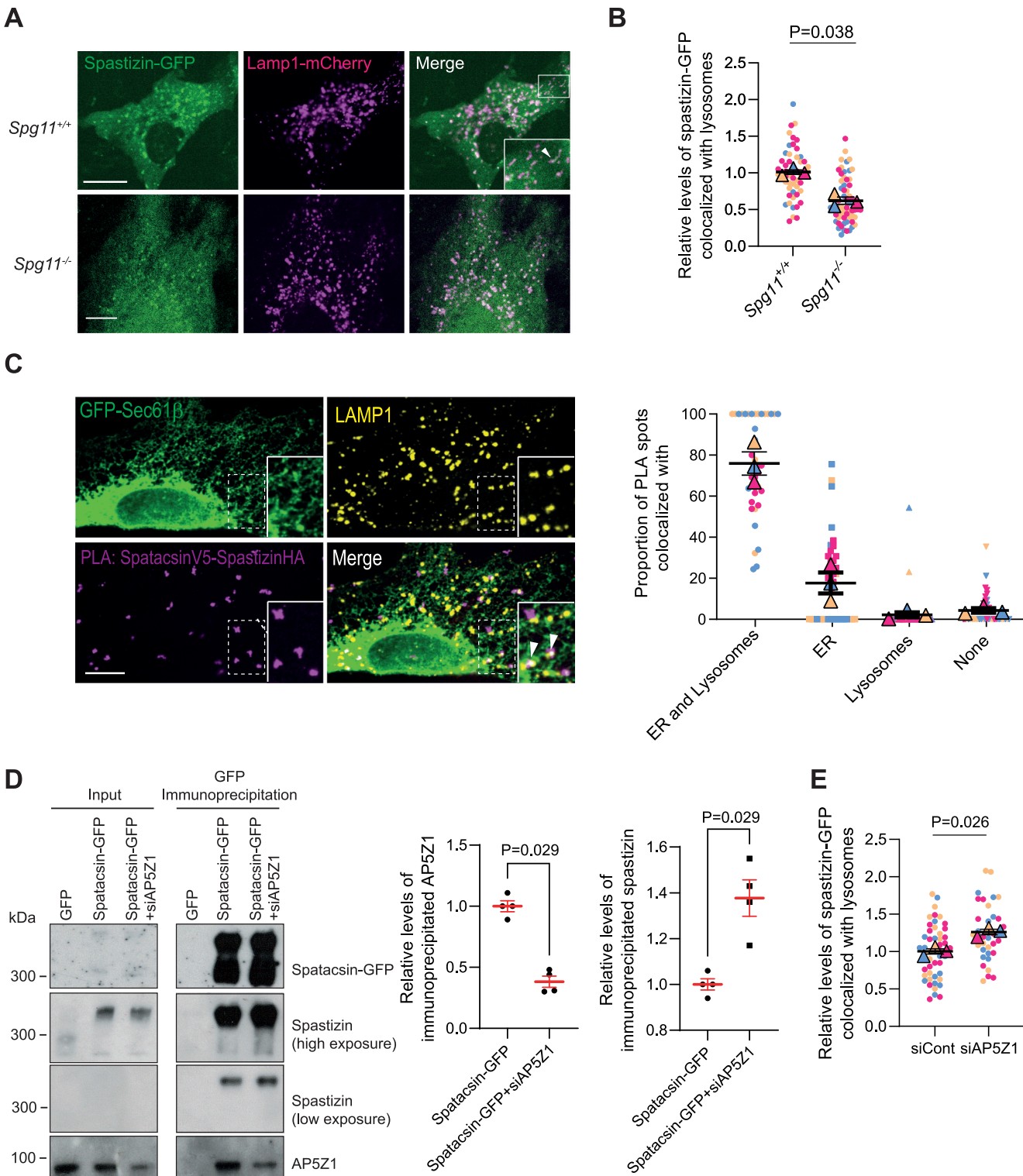

**Fig 6. Spatacsin-mediated degradation of AP5Z1 promotes spastizin recruitment to lysosomes.** (**A**) Expression of spastizin-GFP and Lamp1-mCherry in $Spg11^{+/+}$ and $Spg11^{-/-}$ MEFs. Note the localization of spastizin-GFP along the tubular lysosomes in $Spg11^{+/+}$ MEFs (arrowheads in insert), and the weaker colocalization of spastizin-GFP with lysosomes in $Spg11^{-/-}$ MEFs. Scale bar 10 μm. (**B**) Quantification of the proportion of spastizin-GFP colocalized with Lamp1-mCherry in $Spg11^{+/+}$ and $Spg11^{-/-}$ MEFs. Superplot: means and SEM, $N > 50$ cells from 3 independent experiments. Paired $t$ test on the means. (**C**) Proximity ligation assay (PLA) showing the interaction between V5-tagged spatacsin and HA-tagged spastizin in wild-type MEFs. The PLA signal

(magenta) is detected at the level of the ER labelled by GFP-Sec61β and lysosomes immunostained with Lamp1 (arrowheads). Scale bar: 10 μm. Right: quantification of the proportion of PLA spots colocalized with only the ER, lysosomes, or both the ER and lysosomes. Superplot: means and SEM. $N = 3$ independent experiments. (**D**) Co-immunoprecipitation of spastizin and AP5Z1 with spatacsin-GFP upon down-regulation of AP5Z1 using siRNA (siAP5Z1). Input represents 5% of lysate added to the immunoprecipitation assay. Note that spatacsin and spastizin looked slightly different in input and co-immunoprecipitation, likely due to the high amount of both proteins in immunoprecipitates. Right: quantification of the relative amount of AP5Z1 or spastizin co-immunoprecipitated with Spatacsin-GFP. Means and SEM, $N = 4$ independent experiments. Mann–Whitney test. (**E**) Quantification of the proportion of spastizin-GFP colocalized with Lamp1-mCherry in wild-type MEFs transfected with siRNA down-regulating AP5Z1 (siAP5Z1). Superplot: means and SEM, $N > 52$ cells from 3 independent experiments. Paired $t$ test on the means. The raw data underlying panels B, C, D, and E can be found in S1 Data file.

Spastizin has been shown to interact with the motor protein KIF13A, with an interaction domain mapped close to the motor domain according to two-hybrid analysis [34]. We confirmed by co-immunoprecipitation that KIF13A-YFP and its mutant form devoid of the motor domain KIF13A-ST-YFP [35] were capable of interacting with spastizin (Fig 7D). Of note, the mutant form of KIF13A interacted more with spastizin than wild-type KIF13A. The absence of motor domain in the KIF13A-ST-YFP protein may facilitate access of spastizin to its binding domain. Overexpression of the mutant KIF13A-ST-YFP prevented the formation of tubular lysosomes and altered their dynamics (Fig 7E and 7F), suggesting that KIF13A was important to control the formation of tubular lysosomes and their motility. To test whether interaction of KIF13A with spastizin was important for the formation and motility of tubular lysosomes, we generated a mutant human spastizin lacking the C-terminal domain (aa 2,120 to 2,539) mapped as the interacting domain with KIF13A by two-hybrid analysis [34]. Upon down-regulation of spastizin using siRNA, expression of human wild-type spastizin restored the number of tubular lysosomes and the lysosomal dynamics (S7H and S7I Fig). However, expression of human spastizin lacking its C-terminal domain and unable to interact with KIF13A (S7G Fig) did not restore normal tubular lysosome functions (S7H and S7I Fig). These data showed that the C-terminal domain of spastizin was required to promote the formation and motility of tubular lysosomes, likely by interacting with the motor protein KIF13A.

The formation of tubular membrane organelles requires the coordination of numerous effectors [36]. We therefore investigated whether AP5Z1 might also interact with some motor protein. The endocytic adaptor protein complex AP2 was shown to interact with p150$^{\text{Glued}}$, a subunit of dynein/dynactin complex implicated in retrograde transport [37,38]. Co-immunoprecipitation showed that AP5Z1 also interacted with p150$^{\text{Glued}}$ (Fig 7G). To test whether this protein contributed to lysosome dynamics, we expressed a dominant negative construct p150$^{\text{Glued}}$-CC1 [39], which prevented the formation of tubular lysosomes and altered their dynamics (Fig 7H and 7I). These results suggested that AP5Z1 interaction with p150$^{\text{Glued}}$ contributed to the formation of tubular lysosomes and regulates their motility.

Together, these data show that both spastizin and AP5Z1 contributed to the formation of tubular lysosomes and regulated their trafficking. Both effects on morphology and trafficking of lysosomes were likely mediated by the interactions of spastizin and AP5Z1 with anterograde and retrograde motors proteins, respectively. Since spatacsin controlled the levels of AP5Z1 as well as the association of spastizin with lysosomes, the balance between spastizin and AP5Z1 may regulate the directionality of lysosome movements.

## Spatacsin regulates directionality of lysosome trafficking in axons

We then investigated highly polarized neurons that are degenerating in the absence of spatacsin, spastizin, or AP5Z1 [22,23,29,30] to test whether spatacsin may regulate direction of lysosome trafficking. Primary cultures of wild-type or $Spg11^{-/-}$ cortical neurons were transfected

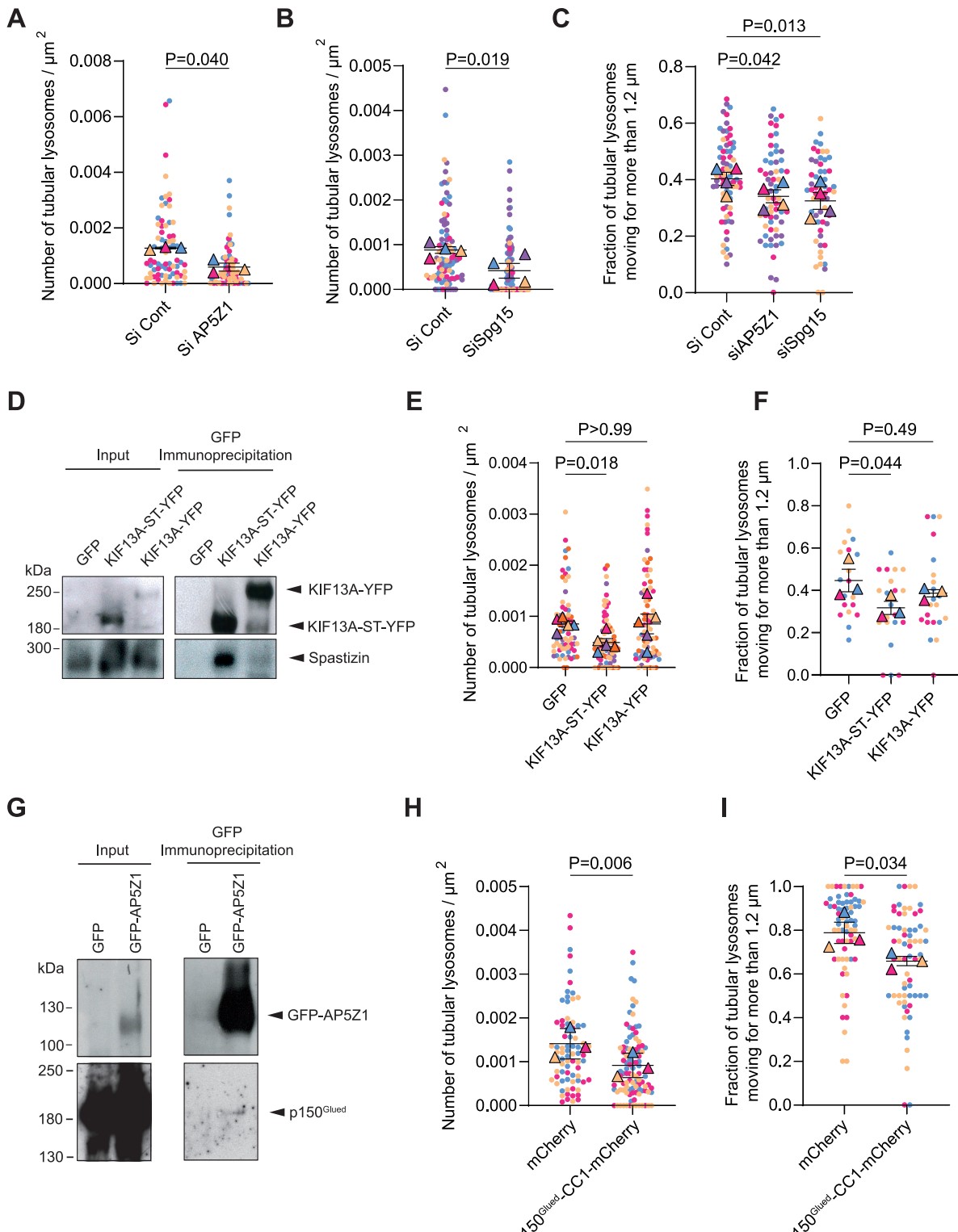

**Fig 7. Spastizin and AP5Z1 regulate association of lysosomes with motor proteins.** (**A**) Quantification of the number of tubular lysosomes in wild-type MEFs transfected with a control siRNA or an siRNA down-regulating AP5Z1. Superplot: means and SEM, $N = 74$ cells from 3 independent experiments. Paired $t$ test on the means. (**B**) Quantification of the number of tubular lysosomes in wild-type MEFs transfected with a control siRNA and an siRNA down-regulating Spg15. Superplot: means and SEM, $N > 92$ cells from 4 independent experiments. Paired $t$ test on the means. (**C**) Proportion of tubular lysosomes moving $> 1.2$ μm over 1 minute in wild-type MEFs transfected

with a control siRNA, an siRNA down-regulating AP5Z1 or Spg15. Superplot: means and SEM, $N > 60$ cells from 3 different independent experiments. RM one-way ANOVA on the means, Dunnett's multiple comparisons test. (**D**) Western blots showing co-immunoprecipitation of spastizin-HA with KIF13A-YFP or mutant KIF13A-ST-YFP (devoid of the motor domain). Input represents 5% of lysate added to the immunoprecipitation assay. (**E**) Quantification of the number of tubular lysosomes in wild-type MEFs transfected with wild-type KIF13A or mutant KIF13A-ST. Superplot: means and SEM, $N > 69$ cells from 4 independent experiments. RM one-way ANOVA on the means, Dunnett's multiple comparisons test. (**F**) Proportion of tubular lysosomes moving $> 1.2$ μm over 1 minute in wild-type MEFs transfected with wild-type KIF13A or mutant KIF13A-ST. Superplot: means and SEM, $N > 19$ cells from 3 different independent experiments. RM one-way ANOVA on the means, Dunnett's multiple comparisons test. (**G**) Western blots showing co-immunoprecipitation of p150$^{Glued}$ with GFP-AP5Z1. Input represents 5% of lysate added to the immunoprecipitation assay. (**H**) Quantification of the number of tubular lysosomes in wild-type MEFs transfected with a vector overexpressing the dominant negative construct p150$^{Glued}$-CC1-mCherry. Superplot: means and SEM, $N > 77$ cells from 3 independent experiments. Paired $t$ test on the means. (**I**) Proportion of tubular lysosomes moving $> 1.2$ μm over 1 minute in wild-type MEFs transfected with the dominant negative construct p150$^{Glued}$-CC1-mCherry. Superplot: means and SEM, $N > 65$ cells from 3 different independent experiments. Paired $t$ test on the means. The raw data underlying panels A, B, C, E, F, H, and I can be found in S1 Data file.

with Lamp1-GFP and mCherry-TRIM46 and analyzed after 7 days in vitro. Since the uniform polarity of microtubules in axons facilitates the analysis of transport [40], the analysis of lysosomal trafficking was focused on axons, which were identified by the presence of the axon initial segment protein TRIM46 (S8A Fig). As observed in MEFs, tubular lysosomes in axons were more motile than the general population of lysosomes (Fig 8A and 8B). Furthermore, as in MEFs, neurons devoid of spatacsin presented a lower number of tubular lysosomes along the axon (Fig 8C), and a lower proportion of lysosomes were engaged in a dynamic movement in absence of spatacsin (Fig 8D and 8E).

Furthermore, investigation of lysosome trafficking in axons allowed us to monitor the direction of their displacement. Analysis of the directionality of movement showed a lower proportion of lysosomes with anterograde movement in $Spg11^{-/-}$ axons compared to wild-type axons, whereas the proportion of lysosomes with retrograde movement was not significantly modified (Figs 8F and S8B). Importantly, axons of wild-type but not $Spg11^{-/-}$ neurons presented similar proportions of lysosomes with anterograde and retrograde movement (Figs 8F and S8B). The imbalance between anterograde and retrograde movement observed in $Spg11^{-/-}$ axons led to a concentration of lysosomes in the proximal part of their axons (Fig 8G). Together, these data suggest that spatacsin regulates the equilibrium between anterograde and retrograde trafficking of lysosomes and therefore affects the distribution of lysosomes along the axon.

We then investigated whether the action of spatacsin on lysosome trafficking in axons may be affected by the relative levels of spatacsin and AP5Z1. Overexpression of GFP-AP5Z1 in neurons expressing Lamp1-mCherry increased the proportion of lysosomes with retrograde movement in axons (S8C and S8D Fig). Conversely, down-regulation of AP5Z1 levels using an siRNA promoted anterograde movement (Fig 8H) but did not modify retrograde movement (Fig 8I). Overall, these results showed that the levels of AP5Z1 may have an action on the directionality of lysosome trafficking in axons. As spatacsin regulates the levels of AP5Z1 by promoting its degradation by lysosomes, this mechanism likely explained the change in the directionality of lysosome trafficking and repartition of lysosomes in the axon of $Spg11^{-/-}$ neurons (Fig 8J).

## Discussion

The loss of spatacsin, involved in hereditary spastic paraplegia type SPG11, causes lysosomal dysfunction [20,22,23,41]. However, the molecular function of spatacsin has, thus far, remained elusive. Here, using biochemical methods and super-resolution STED microscopy, we establish that spatacsin is a protein present in the ER that controls the trafficking of

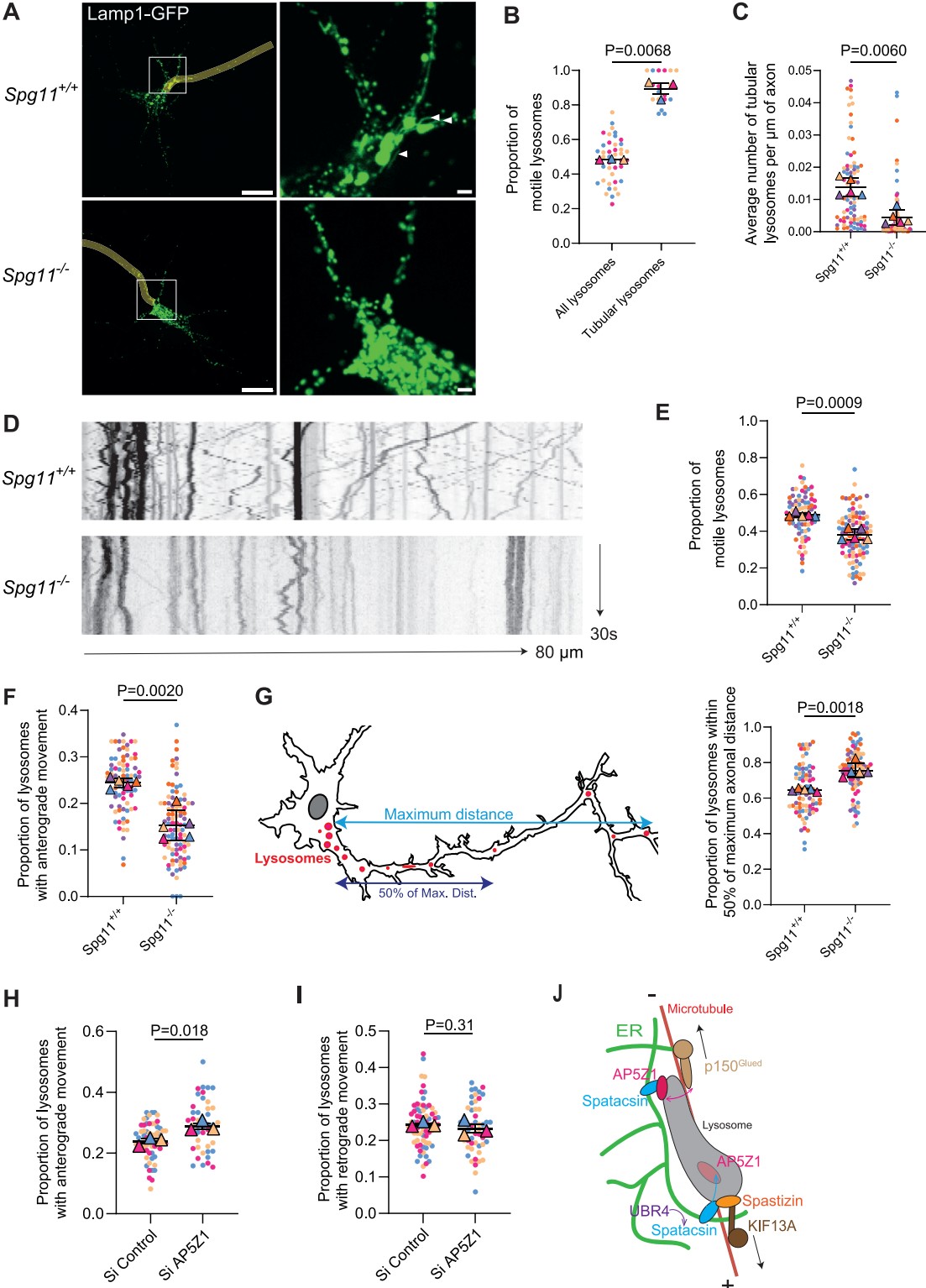

**Fig 8. Spatacsin regulates directionality of lysosome trafficking in axons.** (**A**) Live images of *Spg11*^+/+ and *Spg11*^−/− primary mouse neurons at DIV7 expressing Lamp1-GFP. Axons are highlighted in yellow. White arrows point tubular lysosomes. Note the loss of tubular lysosomes in *Spg11*^−/− neurons. Scale bar: 10 μm, inset: 1 μm. (**B**) Quantification of the proportion of motile lysosomes among the total population of lysosomes or tubular lysosomes in wild-type neurons. Superplot: means and SEM, *N* > 18 neurons from 3 independent experiments. Paired *t* test on the means. (**C**) Quantification of the number of tubular lysosomes along

the axon in $Spg11^{+/+}$ and $Spg11^{-/-}$ mouse neurons. Superplot: means and SEM, $N > 80$ neurons from 5 independent experiments. Paired $t$ test on the means. (**D**) Kymographs representing lysosomes movement over 30 seconds along the axon of $Spg11^{+/+}$ and $Spg11^{-/-}$ primary mouse neurons. Left side of the kymographs is toward the soma, and right is toward the axon. Length of the axon segment: 80 µm. (**E**) Quantification of the proportion of lysosomes that are motile along the axon of $Spg11^{+/+}$ and $Spg11^{-/-}$ primary mouse neurons. Superplot: means and SEM, $N > 79$ neurons from 5 independent experiments. Paired $t$ test on the means. (**F**) Quantification of the proportion of lysosomes that are moving anterogradely along the axon of $Spg11^{+/+}$ and $Spg11^{-/-}$ primary mouse neurons. Superplot: means and SEM, $N > 79$ neurons from 5 independent experiments. Paired $t$ test on the means. (**G**) Left: Scheme representing the distribution of lysosomes along the axon of mouse neurons; the maximum distance considered is from the farthest detected axonal lysosome from the soma to the beginning of the axon. Right: Quantification of the proportion of lysosomes that are found within 50% of the maximum distance along the axon in $Spg11^{+/+}$ and $Spg11^{-/-}$ primary mouse neurons. Superplot: means and SEM, $N > 80$ neurons from 5 independent experiments. Paired $t$ test on the means. (**H**) Quantification of the proportion of lysosomes that are moving anterogradely along the axon of $Spg11^{+/+}$ primary mouse neurons upon transfection with Lamp1-GFP and control siRNA or siRNA down-regulating AP5Z1. Superplot: means and SEM, $N > 50$ neurons from 3 independent experiments. Paired $t$ test on the means. (**I**) Quantification of the proportion of lysosomes that are moving retrogradely along the axon of $Spg11^{+/+}$ primary mouse neurons upon transfection with Lamp1-GFP and control siRNA or siRNA down-regulating AP5Z1. Superplot: means and SEM, $N > 50$ neurons from 3 independent experiments. Paired $t$ test on the means. (**J**) Scheme showing that spatacsin regulates the levels of AP5Z1 by promoting its degradation using a pathway implicating UBR4. This regulates the amount of AP5Z1 and spastizin at the lysosome surface. Since AP5Z1 and spastizin interact with the retrograde p150[Glued] and anterograde KIF13A motor proteins, respectively, the regulation of AP5Z1 and spastizin levels at lysosome surface is a mechanism regulating directionality of lysosome trafficking. The raw data underlying panels B, C, E, F, G, H, and I can be found in S1 Data file.

lysosomes. Using a combination of trained neural network and targeted image analysis coupled to an siRNA screen, we show that spatacsin function is associated to protein degradation pathways. We demonstrate that spatacsin controls the directionality of lysosome movement by modulating the degradation of AP5Z1.

The subcellular localization of spatacsin has long been debated [17,18] and is still unclear, notably due to the lack of specific antibody to detect the endogenous protein. Our biochemical experiments suggest that spatacsin is tightly attached to membranes, likely by transmembrane regions [16], and that it is present in the light membrane fractions containing ER proteins, as well as the late endosome and lysosome marker Lamp1, but not the lysosomal hydrolase cathepsin D. Such discrepancy between Lamp1 and Cathepsin D distribution has been observed by others [42], and it suggests that spatacsin is poorly associated with degradative lysosomes. To discriminate spatacsin localization between nondegradative Lamp1-positive compartment and the ER, we relied on subcellular localization using overexpression of tagged spatacsin. Our data with STED microscopy support that spatacsin is localized in the ER compartment. Even if we chose for our analysis cells with low levels of spatacsin overexpression, we have to keep in mind that overexpression may affect the localization of the protein.

Independently of its localization, the absence of spatacsin slightly impaired the contacts between the ER and lysosomes. Contacts between the ER and lysosomes are mediated by a variety of molecular actors and play a role in numerous functions such as regulation of calcium homeostasis, cholesterol transfer between compartments, or regulation of lysosomes positioning or dynamics [6,7]. Several molecular factors regulating contacts between the ER and lysosomes restrict lysosome dynamics. For example, SNX19 or the ubiquitination of p62 by the ER-localized ubiquitin ligase RNF26 maintain lysosomes in the perinuclear region of cells [9,43]. When cholesterol content in lysosome membrane is low, the Rab7 effector ORP1L adopts a conformation allowing its interaction with the ER protein VAP, maintaining lysosomes in the cellular periphery [11]. In contrast, we show here that spatacsin promotes contacts between the ER and lysosomes and regulates lysosome motility, highlighting the diversity of functions controlled by contact sites between the ER and lysosomes.

The importance of spatacsin in lysosome function has been largely documented [20–23,41] as revealed by its impact on tubular lysosomes. The formation of tubules emanating from

lysosomes has also been observed after long-term starvation in autophagic lysosome reformation (ALR), a cellular process that relies on spatacsin, spastizin, and AP5Z1, although the implication of the latter is debated [20,29]. Starvation has been shown to promote lysosomal recruitment of spastizin in a Rag GTPase-dependent manner [27]. The recruitment of spastizin to lysosomes may contribute to the formation of tubules in ALR. However, the tubular lysosomes observed in our study without induction of starvation are catalytically active lysosomes, whereas ALR tubules are catalytically inactive [44]. Tubular lysosomes with different properties may be associated with different cellular functions [2], but it would be interesting to investigate whether the mechanisms highlighted in our study are also valid for ALR-mediated lysosomal tubules.

The tubular lysosomes observed by live imaging both in mouse embryonic fibroblasts and neurons in control conditions were more motile than round lysosomes, as observed in other cell types [45]. All experimental conditions decreasing the proportion of tubular lysosomes also impaired their dynamics, suggesting that the formation of tubular lysosomes is associated with their trafficking. Formation of tubular lysosomes and lysosome trafficking were previously associated with the kinesin KIF5B [46,47]. Here, we demonstrate the importance of another kinesin, KIF13A, or the dynein/dynactin subunit p150$^{Glued}$. These motor proteins likely contribute to the formation of tubular lysosomes by pulling on membranes, as shown in endosomes [48,49]. The motor proteins that associate with lysosomes to promote their tubular shape may also contribute to their trafficking, which would explain why tubular lysosomes are overall more dynamic. The higher traffic observed for tubular lysosomes may thus be a downstream consequence of their morphology caused by recruitment of motor proteins to these lysosomes.

The motor proteins KIF13A and p150$^{Glued}$ are likely associated to lysosomes by spastizin and AP5Z1, respectively. Interaction of KIF13A with spastizin is likely direct, as their interaction was identified by two-hybrid screen and confirmed by GST pulldown [34] and co-immunoprecipitation in our study. The interaction of spastizin with KIF13A, which is a plus end-directed microtubule motor allows the movement of vesicles toward the cell periphery [50]. In contrast, AP5Z1 interaction with the subunit of dynein/dynactin p150$^{Glued}$ likely promotes the movement of lysosomes toward the minus-end of microtubules [38]. Of note, the interaction of AP5Z1 with p150$^{Glued}$ may be indirect and could occur through another subunit of the AP5 complex, similar to what has been observed for adaptor protein complex AP2 that interacts with p150$^{Glued}$ via its β subunit [37]. Our data show that spatacsin regulates the association of spastizin and AP5Z1 with the membrane of lysosomes. The association of spastizin with lysosomes requires its interaction with spatacsin likely at the level of contact sites between the ER and lysosomes. Importantly, the levels of AP5Z1 appear to regulate the association of spastizin with lysosomes. AP5Z1 levels must be tightly regulated as both its overexpression and downregulation impaired tubular lysosome formation and dynamics. This might be explained by the fact that formation of a tubular organelles likely requires coordination of motor proteins with opposite action [48,49]. The degradation of AP5Z1 mediated by spatacsin could thus be a mechanism to control the relative amount of both spastizin and AP5Z1 at the lysosomal surface. Since AP5Z1 and spastizin interact with a minus-end- or a plus-end-directed motor protein, respectively, spatacsin likely regulates the formation of tubular lysosomes and orientates the direction of lysosome movement by modulating the equilibrium between AP5Z1 and spastizin at the lysosome surface.

This function of spatacsin is confirmed by the lower anterograde trafficking of lysosomes observed in axons of $Spg11^{-/-}$ neurons. A similar phenotype was observed in neurons devoid of spastizin [51]. Furthermore, the coupling of motor proteins to lysosomes mediated by spatacsin may also contribute to the distribution of lysosomes in other cell types and may explain

the clustering of lysosomes around the nucleus observed in fibroblasts when spatacsin or spastizin are absent [19,51].

The imbalance between retrograde and anterograde transport in $Spg11^{-/-}$ axons resulted in the enrichment of lysosomes toward the neuronal soma while the axons were slightly deprived of lysosomes. Several recent publications have highlighted the importance of axonal lysosomal trafficking in the development of neurodegenerative disease [5,52]. In SPG11 models, previous work has shown that cultured neurons deprived of spatacsin presented axonal instability [53]. Therefore, the impaired distribution of lysosomes along the axon in absence of spatacsin, which results of impaired lysosomal dynamics, appears to be particularly relevant to the development of hereditary spastic paraplegia type 11.

The regulation of lysosome trafficking by spatacsin relies on the tight regulation of AP5Z1 levels, as both its overexpression and down-regulation impaired the dynamics of lysosomes both in MEFs and axons of neurons. Spatacsin promotes AP5Z1 degradation by lysosomes. Interestingly, overexpression of spatacsin also promoted degradation of another subunit of the AP5 complex, AP5M1, suggesting that spatacsin may regulate the degradation of the whole complex. Spatacsin likely promoted monoubiquitination of AP5Z1 that has been associated with lysosomal degradation in few examples [54–56]. However, ubiquitinated AP5Z1 has an apparent molecular weight lower than the endogenous protein, suggesting that it may undergo some processing. Of note, spatacsin-independent pathways may also exist to promote the degradation of AP5Z1, as the levels of AP5Z1 have been shown to be strongly decreased in fibroblasts of SPG11 and SPG15 patients [57]. It was proposed that AP5Z1 stability may depend on the presence of spatacsin and spastizin [18]. Therefore, spatacsin may contribute to tightly regulate AP5Z1 levels by contributing both to its stability and to its degradation.

The degradation of AP5Z1 by spatacsin is partially prevented by UBR4 down-regulation. However, the lack of a known ubiquitination domain in UBR4 [58] suggests an indirect action of UBR4 on AP5Z1 degradation. Our data suggest that UBR4 contributes to stabilize spatacsin. The interaction of spatacsin with UBR4 detected by the two-hybrid screen and by co-immunoprecipitation suggests that UBR4 may act on spatacsin stabilization, which would also affect AP5Z1 levels. This highlights that levels of spatacsin and AP5Z1 are tightly regulated and interdependent. Consistently, spatacsin also appears to be regulated by ubiquitination upon overexpression. Our screening strategy identified several proteins likely associated with degradation pathways as regulators of spatacsin function. Whether the other spatacsin interactors we identified directly act on spatacsin or may modulate the levels of spatacsin partners such as AP5Z1 remains to be elucidated. It can be speculated that these interactors may contribute to finely regulate the levels of spatacsin and its partners, which likely affects the formation of tubular lysosomes and their trafficking.

In conclusion, we identify spatacsin as a protein present in the ER that regulates the trafficking of lysosomes. We demonstrate that spatacsin, by promoting the lysosomal degradation of AP5Z1, regulates the lysosomal association of AP5Z1 and spastizin. The latter are binding motor proteins with antagonistic role regarding lysosome dynamics. The control of AP5Z1 degradation by spatacsin thus appears as a mechanism to regulate the directionality of lysosome movement, which likely has an impact on lysosome distribution in highly polarized cells such as neurons.

## Experimental procedures

**Ethics statement.**   The care and treatment of animals followed the N° 2010/63/UE European legislation and national authority (Ministère de l'agriculture, France) guidelines for the detention, use, and ethical treatment of laboratory animals. All the experiments were approved

by ethics committee (Comité d'éthique en experimentation animale N˚005) and the French Ministère de l'Agriculture (#5199 201604201549915 approval number). All experiments were conducted by authorized personnel.

## Mouse models

The Spg11 knockout ($Spg11^{-/-}$) model has been previously described [22]. It was generated by inserting 2 stop codons in exon 32, leading to the loss of expression of spatacsin, and can thus be considered as a functional Spg11 knockout. To obtain this mouse model, we created an intermediate model in which floxed exons 32 to 34 bearing the stop codons were inserted in the reverse orientation in intron 34 [22] (S3A Fig). Reverse transcription PCR (RT-PCR) of the transcripts followed by sequencing of brain and spleen samples showed this intermediate model to express the floxed allele, with the splicing of exons 32 to 34 and conservation of the reading frame between exon 31 and exon 35 (S3B and S3C Fig). It was thus equivalent to a functional deletion of exons 32 to 34 and was named $Spg11^{\Delta 32-34/\Delta\ 32-34}$. Behavioral evaluation of the mice and immunohistochemical analysis of the mouse brains were performed as previously described [22].

## Antibodies

The antibodies used for immunofluorescence and immunoblotting were rat anti-Lamp1 (clone 1D4B, Development Studies Hybridoma Bank, University of Iowa, USA, deposited by JT August), rabbit anti-V5 (Cat#8137, Sigma), mouse anti-V5 (Cat#Ab27671, Abcam), rat anti-HA (clone 3F10, Cat#11867423001, Merck), rabbit anti-HA (Cat#ab9110, Abcam), rabbit anti-GFP (Cat#6556, Abcam), mouse anti-Myc (clone 9E10, Development Studies Hybridoma Bank), rabbit anti-Stim1 (Cat#5668, Cell Signaling Technology), rabbit anti-cathepsin D (Cat#Ab75852, Abcam), rabbit anti-spatacsin (Cat#16555-1-AP, ProteinTech), rabbit anti-spastizin (Cat#5023, ProSci), rabbit anti-AP5Z1 (Cat#HPA035693, Sigma), rabbit anti AP5M1 (provided by Dr. J Hirst; [18]), mouse anti-clathrin heavy chain (clone 23, Cat#610500, BD Biosciences), rabbit anti-UBR4 (Cat#Ab86738, Abcam), mouse anti-p150$^{Glued}$ (Cat#610474, BD Biosciences), rabbit anti-calreticulin (Cat#SPA600F, Enzo Life Sciences), mouse anti-VDAC1 (Cat#Ab16814, Abcam), rabbit anti-REEP5 (14643-1-AP, ProteinTech), mouse anti-p62 (Cat#Ab 56416, Abcam), rat anti Lamp2 (Cat#Ab13524, Abcam), mouse anti-α-tubulin (Clone DM1A, Cat#Ab7291, Abcam), mouse anti-actin (Clone C4, Cat#Ab3280, Abcam), and mouse anti-ubiquitine (Clone P4D1, Cat #3936S, Cell Signaling Technology).

The secondary antibodies used for immunofluorescence were purchased from Thermofisher: donkey anti-mouse IgG Alexa 488 (Cat#A21202), goat anti-rabbit IgG Alexa 555 (Cat#A21429), and goat anti-rat IgG Alexa 647 (Cat#A21247). For STED microcopy, the secondary antibodies were anti-rabbit Atto-488 (Cat#A11008), anti-mouse IgG2a Alexa-594 (Cat#A21135), and goat anti-rat Alexa-647 (Cat#A21247). The secondary antibodies coupled to horseradish peroxidase used for immunoblotting were purchased from Jackson ImmunoResearch (Ely, UK): donkey anti-mouse IgG (Cat#JIR715-035-151) and donkey anti-rabbit IgG (Cat#**711-035-152**).

## Plasmids

The human spastizin-GFP and spastizin-V5 vectors have been previously described [25]. Spastizin-HA was obtained by replacing GFP with an HA tag in the spastizin-GFP vector. To generate spastizinΔCT-GFP and spastizinΔCT-V5 vectors, we amplified by PCR (using Platinum SuperFi II PCR Master Mix, Thermofisher) the fragment of human *Spg15* encoding aa 1 to 2,120 and used Gibson cloning kit (New England Biolabs) to replace *Spg15* cDNA in the

spastizin-GFP or spastizin-V5 vectors. A codon-optimized vector expressing human spatacsin was generated (Baseclear, Leiden, the Netherlands) in a gateway compatible system (Thermofisher). The cDNA was transferred by LR clonase into the pDest-47 vector or pDest-53 (Thermofisher), leading to a vector expressing spatacsin in fusion with a C-terminal or N-terminal GFP, respectively. Deletion of nucleotides 6,013 to 6,477 of optimized spatacsin cDNA resulted in spatacsin$^{\Delta 32-34}$. C-terminal fragments of spatacsin (aa 1,943 to 2,433) and spatacsin$^{\Delta 32-34}$ (aa 1,943 to 2,226) were amplified by PCR and inserted in the pDest-53 vector (Thermofisher), leading to vectors that expressed GFP-spatacsin-Cter and GFP-spatacsin-Cter$^{\Delta 32-34}$. Vector expressing AP5Z1-His was obtained from M. Slabicki [26]. To generate GFP-AP5Z1 or GFP-mCherry-AP5Z1 constructs, a stop codon was inserted after the final codon of AP5Z1 in the AP5Z1-His vector, and AP5Z1 cDNA was transferred by LR clonase into the pDest-53 or the pDEST-CMV-N-Tandem-mCherry-EGFP vector (Addgene #123216), respectively. The other plasmids used in the study were obtained from other laboratories or Addgene. GFP-Sec61β was obtained from G. Voeltz [59], reticulon2-V5 was from E. Reid [60], KIF13A-YFP and KIF13A-ST-YFP were from C. Delevoye [35], p150$^{Glued}$-CC1-mCherry was from T. Schroer [39], Ubiquitin-6His was obtained from R. Baer [61], Lamp1-GFP (#16290), Lamp1-mCherry (#45147), and mCherry TRIM46 (#176401) were from Addgene.

## siRNA

The siRNAs used to down-regulate *Spg11* were either On-target plus siRNAs (Dharmacon), with the sequences CAGCAGAGAGUUACGCCAA (#J-047107-09-0002) and CAGUAUGUGCCGGGAGAUA (#J-047107-12-0002), or from Thermofisher, with the sequence GGUUCUACCAGGCUUCUAUtt (#s103130). The siRNAs used to down-regulate *AP5Z1* and *Spg15* were Silencer Select siRNAs form Thermofisher: GGAGCAGAGUAACCGGAGAtt (#s106997, *AP5Z1*) and UCUGCUCCCGGGUCACUAAtt (#s106999, *AP5Z1*), CUUCAACUCCUGCAACGAAtt (#s102537, *Spg15*) and GAGCGAUACCAAGAGGUAAtt (#s102536, *Spg15*). The siRNAs used to test the role of spatacsin interactors identified by the two-hybrid screen were Silencer Select siRNAs from ThermoFisher and are listed in S5 Table.

## Subcellular fractionation of brain tissue

Mice were killed using $CO_2$ and the brains immediately dissected and rinsed twice in PBS at 4°C. Subcellular fractionation was performed according to a previously described procedure [31]. Dissected brains were homogenized in 0.32 M sucrose and 10 mM HEPES (pH 7.4) using a PFTE (polytetrafluoroethylen) pestle attached to a stirrer (Heidolph, Germany) rotating at 500 rpm. Lysates were centrifuged at 1,330 × *g* for 3 minutes, generating a pellet (P1) and a supernatant (S1). The S1 supernatant was centrifuged at 21,200 × *g* for 10 minutes, producing a pellet (P2) and a supernatant (S2). The S2 supernatant was then centrifuged at 200,000 × *g* for 1 hour, generating a pellet (P3) and a supernatant (S3).

## Membrane association assay

To determine the membrane association of spatacsin, mouse brain were extracted and homogenized as described above. After a first centrifugation at 1,330 × g for 3 minutes, the post-nuclear supernatant was centrifuged at 200,000 × g for 1 hour to collect a pellet corresponding to membranes. To evaluate spatacsin association with the membrane fraction, the latter was processed as previously described [31]. The membrane fraction was treated with one of the following solutions: 1 M NaCl and 25 mM phosphate buffer (pH 7.4); 100 mM glycine buffer (pH 2.8); 100 mM carbonate buffer (pH 11.0); or 1.0% sodium deoxycholate and centrifuged at

$200,000 \times g$ for 60 minutes. The final pellet and the supernatants were analyzed by western blotting.

## Isolation of the ER- and lysosome-enriched fractions

Isolation of the ER- and lysosome-enriched fractions was performed according to previously described protocols with several modifications [62,63]. After killing mice using $CO_2$, the brains were immediately extracted and washed with PBS at 4°C. Brains were mechanically dissociated in 250 mM sucrose, 1 mM EDTA, 10 mM HEPES (pH 7.4), 1 mM DTT, and 25 mM KCl supplemented with a protease inhibitor cocktail (Thermofisher), using a PFTE pestle attached to a stirrer (Heidolph, Germany) rotating at 500 rpm. Lysates were centrifuged at $800 \times g$ for 5 minutes and the pellets discarded. The supernatant was centrifuged at $20,000 \times g$ for 10 minutes and the resulting pellet A retained for lysosome isolation and the supernatant A for ER isolation. For lysosome isolation, pellet A was resuspended in 2 mL of the initial buffer and deposited on 10 ml of 27% Percoll solution in a 15-mL tube. After 90 minutes of centrifugation at $20,000 \times g$, the lysosomal fraction was visible close to the bottom of the tube and collected by pipetting. It was then resuspended in the initial buffer and centrifuged at $20,000 \times g$ for 10 minutes. The pellet was resuspended in sample buffer and analyzed by western blotting.

To isolate the ER, supernatant A was deposited on a gradient of several sucrose solutions prepared in 10 mM Tris (pH 7.4) and 0.1 mM EDTA. The sucrose concentrations of the 3 solutions were from the bottom up: 2 M, 1.5 M, and 1.3 M. The preparation was centrifuged for 70 minutes at $152,000 \times g$. After centrifugation, the ER-enriched fraction was found at the phase limit between the 1.3 M sucrose solution and the 1.5 M sucrose solution. The fraction was collected and resuspended in the initial buffer and centrifuged for 45 minutes at $152,000 \times g$. The pellet was resuspended in sample buffer and analyzed by western blotting.

## Mouse embryonic fibroblast cultures

Mouse embryonic fibroblasts were prepared using 14.5-day-old embryos obtained from the breeding of heterozygous ($Spg11^{+/-}$ or $Spg11^{+/\Delta32-34}$) mice as previously described [19]. Comparisons between mutant and wild-type fibroblasts were always performed using fibroblasts originating from embryos of the same breeding. All experiments were performed with fibroblasts between passages 4 and 6.

## Transfection of fibroblasts

Fibroblasts were transfected using the NEON transfection system (Thermofisher) with 1 pulse of 30 ms at 1,350 V, according to manufacturer instructions. Cells ($5 \times 10^5$) were transfected with 5 μg plasmid and analyzed 24 hours later. When we cotransfected a vector expressing a fluorescent protein together with a vector expressing a nonfluorescent protein for live imaging, we imaged cells expressing the fluorescent protein and then fixed the cells afterwards to verify that >95% of cells expressing the fluorescent marker were also positive for the nonfluorescent protein by immunostaining. For transfection with siRNA, $50 \times 10^3$ cells were transfected with 1 pmol siRNA and analyzed after 48 hours in culture.

## Primary cultures of neurons and transfections

Mouse cortical neurons were prepared using cortices of 14.5-day-old embryos obtained from the breeding of heterozygous ($Spg11^{+/-}$) mice. After a chemical dissociation for 15 minutes at 37°C (Trypsin 0.05% in Hibernate medium, Thermofisher) and mechanical dissociation, cortices were passed through a 70-μm filter and plated at a density of 100,000 neurons per $cm^2$ on

precoated with Poly-L-Lysine (100 mg/L) 8-well coverslip IBIDI chambers. Neurons were grown in Neurobasal medium supplemented with B27 (ThermoFisher). Neurons were transfected on day 4 of culture. For each well of the IBIDI chamber, we prepared a mix containing 0.6 μl of Lipofectamine 2000 in 15 μl Opti-Mem medium (Thermofisher), with 500 ng of DNA or 1 pmol of siRNA that was added to cultured neurons for 3 hours.

## Chemicals

Lysotracker Blue, Green, and Red (Thermofisher) were used at 50 nM for 30 minutes to stain acidic lysosomes in fibroblasts. DQ-Red-BSA and DQ-Green-BSA (Thermofisher) were added to the culture medium at 2 μg/ml 1 hour before imaging and then washed once with culture medium. Texas Red–conjugated dextran (10,000 MW, Thermofisher) was added to the culture medium at 100 μg/ml and the cells incubated for 4 hours to allow its internalization by endocytosis and chased for 24 hours to stain the lysosomal compartment. The PI3 kinase inhibitor wortmannin (Sigma) was used at 100 nM for 1 hour. The proteasome inhibitor MG132 was purchased from Tocris and was added to cells for 16 hours at 15 μM. Bafilomycin (Tocris) was added to cells for 16 hours at 100 nM.

## Immunofluorescence

Cells were fixed in 4% PFA in PBS for 20 minutes and then permeabilized for 5 minutes in PBS containing 0.2% v/v Triton X-100. Cells were then blocked for 45 minutes in PBS with 5% w/v BSA (PBS-BSA) and incubated with primary antibodies in PBS-BSA overnight at 4°C. Cells were washed 3 times with PBS and incubated with secondary antibodies coupled to fluorophores. After 3 washes with PBS, glass coverslips were then mounted on glass slides using Prolong Gold antifade reagent (Thermofisher).

## Confocal microscopy

Images of immunofluorescence were acquired using an inverted laser scanning Leica SP8 confocal microscope (Mannheim, Germany) with a 63× objective N.A. 1.40. STED images were acquired on SMD detectors with a confocal laser scanning microscope LEICA SP8 STED 3DX equipped with a 93×/1.3 NA glycerol immersion objective, in a thermostated chamber maintained at 22°C. The specimens were imaged with a white light laser and a pulsed 775 nm depletion laser to acquire nanoscale imaging. Typically, images of 1,024 × 1,024 pixels were acquired with a magnification above 3, resulting in a pixel size in the range of 25 to 45 nm.

For live imaging, cells were imaged at 37°C and 5% $CO_2$ using a Leica DMi8 inverted spinning disk confocal microscope equipped with 63× objective N.A. 1.40 and a Hamamatsu Orca flash 4.0 camera. Timelapses of MEFs were acquired to analyze the trajectories of the lysosomes with 1 image taken every 1 second for 1 minute. For timelapses capturing the trajectories of lysosomes in neurons, 1 image was taken every 500 ms for 30 seconds. Axons were identified by the accumulation of the axon initial segment protein TRIM46 upon transfection of the vector expressing mCherry-TRIM46.

## Electron microscopy

Cultured MEFs were fixed with 2.5% glutaraldehyde in PBS for 2 hours at 22°C. The cells were then postfixed in 1% osmium tetroxide for 20 minutes, rinsed in distilled $H_2O$, dehydrated in 50% and 70% ethanol, incubated in 1% uranyl acetate for 30 minutes, processed in graded dilutions of ethanol (95% to 100%, 5 minutes each), and embedded in Epon. Ultrathin (70 nm)

sections were cut, stained with uranyl acetate and lead citrate, and analyzed with a JEOL 1200EX II electron microscope at 80 kV.

## Two-hybrid screen

The yeast two-hybrid screen was performed by Hybrigenics (Paris, France) using an adult human cDNA brain library. The bait was either the complete 1,943 to 2,443 domain of human spatacsin or the same domain lacking the amino acids encoded by exons 32 to 34.

## Image analysis

**Quantification of the amount of Spatacsin-V5 dots colocalizing with lysosomes and ER.** Spatacsin-V5 and Lamp1 immunostainings of STED images were binarized using the *Spot detector* plugin from ICY. The ER was binarized using Fiji thresholding tool. Then, the amount of Spatacsin-V5 dots that colocalized with the Lamp1 dots or with the ER network was measured using Fiji *Analyse particles* function on the spatacsin dots projected on the mask of the second staining.

**Tubular lysosome detection.** After detecting lysosomal particles using ICY *Spot detector*, the *regionprops* function of the MATLAB Image Processing Toolbox was used to determine the shape characteristics of the particles on the binary images. Tubular lysosomes were defined as follows: circularity $< 0.5$, eccentricity $> 0.9$, and a width/length ratio $> 4$. The selected particles were saved in a new image. To screen the effect of siRNAs on tubular lysosomes, we defined a tubulation index, for which we normalized the number of tubular lysosomes/$\mu m^2$ to a value ranging from 0 to 1, corresponding to the average number of tubular lysosomes/$\mu m^2$ quantified in $Spg11^{-/-}$ and $Spg11^{+/+}$ fibroblasts analyzed in the same experiment.

**Lysosome trajectory analysis.** Analysis of the movement and trajectories of lysosomes over a minute was performed on timelapse images acquired every second. First, to analyze the trajectory characteristics of round and tubular lysosomes particles, the particles were labeled by hand using the multipoint tool of Fiji software to extract their coordinates. The length of the trajectory and the mean speed were then computed using MATLAB. We then performed automated analysis solely for tubular lysosomes using MATLAB software. The *regionprops* function was used to detect the position of the centroids of tubular lysosomes during the timelapse. Once the coordinates were obtained, they were analyzed using John C. Crocker *track. pro* "freeware" MATLAB function to determine the characteristics of the particle trajectories, considering that the maximum theoretical displacement of a particle between 2 frames was the approximate size of 1 tubular lysosome, hence 2.4 μm– 20 pixels. Then, the total distance that each particle traveled was calculated.

**Measurement of the area of the ER–lysosome overlap.** The binarization of lysosomal staining was performed using Spot Detector. The binarization of ER staining was performed using the ImageJ thresholding tool. Once the 2 binary images were obtained, they were compared using MATLAB and the area of overlap between the 2 stainings per particle was measured using the *regionprops* function. We calculated a threshold of overlap consistent with a contact site between ER and a lysosome. Considering a 500-nm wide lysosome within 10 nm of distance of a 50-nm wide ER tubule and the fact that the pixel size for the spinning disk at 63× objective is 120 nm, the overlap of the apparent lysosome (740 nm diameter) and the apparent ER tubule is equal to 230 nm, which represents about 30% of the diameter of the lysosomal staining. Therefore, we chose 30% area overlap between the lysosomal and ER stainings as a threshold for the existence of a contact.

**Analysis of movement directionality using Kymographs.** Kymographs of the trajectories of particles along the axons of mouse neurons were obtained in Fiji, using the *Kymograph Builder* plugin. Once the Kymographs were obtained, the proportion of anterograde/ retrograde and stationary movement of lysosomes was identified by hand.

## Image classification by a neural network

Lysosomes of $Spg11^{+/+}$ and $Spg11^{-/-}$ fibroblasts, as well as $Spg11^{+/+}$ fibroblasts transfected with siRNAs, were stained using Texas Red–conjugated dextran. Images were acquired using a spinning disk confocal microscopy, generating an image library.

To train the Spg11 classification model, Tensorflow (https://www.tensorflow.org/?hl=fr) and Scikit-learn (https://scikit-learn.org/stable/) Python libraries were used. Images of lysosomes of $Spg11^{-/-}$ ($n = 742$ cells) and $Spg11^{+/+}$ ($n = 735$ cells) MEFs were used as a database. Training and test sets were generated randomly with a test set size of 15% (111 $Spg11^{+/+}$ fibroblast images and 112 $Spg11^{-/-}$ fibroblast images). Initial images of $921 \times 1{,}024$ pixels were resized to $224 \times 224$ pixels to reduce input size while retaining consistent information. Data augmentation was performed using flip Tensorflow functions to improve training and artificially increase the number of images. Finally, the image pixel values were normalized between 0 and 1. The transfer learning approach was used to avoid model training from scratch. The VGG16 [64] neural network structure was used and downloaded using the Tensorflow_hub library (https://www.tensorflow.org/hub?hl=fr). VGG16 is a convolutional neural network model trained on ImageNet, which is a dataset of over 14 million images belonging to 1,000 classes. The top 3 layers were excluded and replaced with 3 other layers: one dense layer of 512 neurons, a dropout layer, and a 64-neuron layer. The final layer was the soft-max layer. The neural network was trained using a NVIDIA GeForce GTX 1050 Ti for 150 epochs, with a starting learning rate at 0.0001 and a batch size at 32. Model evaluation resulted in 79.5% total accuracy on the test set.

The trained model was used to predict the probability of the cell to be considered as a $Spg11^{-/-}$ fibroblast for each image of fibroblast transfected with siRNA. For each siRNA, the arithmetic mean of the probability was calculated.

## Protein extraction from cells

MEFs were washed twice with PBS and lysed in 100 mM NaCl, 20 mM Tris (pH 7.4), 2 mM $MgCl_2$, 1% SDS, and 0.1% Benzonase (Sigma). Samples were centrifuged at $17{,}000 \times g$ for 15 minutes and the supernatants recovered as solubilized proteins. The protein concentration was determined using the BCA assay kit (Thermofisher). To evaluate stability of proteins, MEFs were treated with the protein translation inhibitor cylcoheximide (10 μg/ml) for 2 or 4 hours before cell lysis.

## Protein ubiquitination

To detect ubiquitinated proteins, MEFs were transfected with a vector expressing Ubiquitin tagged with a 6-His tag as previously described [65]. Cells were wahed in PBS, lysed in 8 M Urea, 300 mM NaCl, 50 mM $Na_2HPO_4$, 50 mM Tris (pH 7.4), 1 mM $MgCl_2$, 0.5% NP40, 0.1% Benzonase, and 5 mM imidazole. 6-His ubiquitin modified proteins were purified by incubation with NiNTA agarose beads for 1 hour at 22°C. Beads were washed 5 times with lysis buffer containing 15 mM imidazole. Ubiquitin-conjugated proteins were eluted in sample buffer supplemented with 100 mM imidazole and analyzed by western blot.

## Western blotting

Protein lysates supplemented with sample buffer (final concentration 80 mM Tris-HCl (pH 6.8), 10 mM DTT, 2% SDS, 10% glycerol) were separated on 3% to 8% Tris-acetate or 4 to 12 Bis-Tris gels (Thermofisher). Proteins were then transferred to PVDF membranes (Merck). Membranes were then incubated in Ponceau red for 5 minutes and blocked in PBS-0.05% Tween (PBST) with 5% milk for 45 minutes. The membranes were incubated with primary antibodies in PBST-5% milk overnight at 4˚C. Secondary antibodies were conjugated with HRP (Jackson Lab) and the signals visualized using chemiluminescent substrates (SuperSignal West Dura/Femto; Thermofisher). The chemiluminescent signal was then acquired on Amersham Hyperfilm ECL. Images shown in figures are representative of at least 3 independent experiments.

## Co-immunoprecipitation

Cells were lysed on ice in 100 mM NaCl, 20 mM Tris (pH 7.4), 1 mM $MgCl_2$, and 0.1% NP40 supplemented with a protease inhibitor cocktail. Samples were centrifuged at $17,000 \times g$ for 15 minutes at 4˚C. Approximately 5% of the supernatant was retained, supplemented with sample buffer, and was used to monitor protein quantity for the inputs. The remaining 95% of the supernatants was incubated with 10 μL of GFP-trap beads (Chromotek, Germany) for 90 minutes using a rotating wheel at 4˚C. Beads were washed 4 times in lysis buffer and supplemented with sample buffer with DDT. For co-imunoprecipitation of spastizin with spatacsin, cells were transfected with a vector expressing tagged spastizin, as endogenous spastizin was not detected with the chosen lysis buffer. For co-immunoprecipitation of p150[Glued] by AP5Z1 constructs, beads were washed 4 times in 300 mM NaCl, 20 mM Tris (pH 7.4), 1 mM $MgCl_2$, and 0.1% NP40. Beads and inputs were then analyzed by western blotting.

## Proximity ligation assay

MEFs were fixed in PBS containing 4% PFA for 15 minutes. The Duolink Proximity Ligation Assay (PLA DuoLink in situ Starter Kit red, Sigma) was then performed according to the manufacturer's instructions. After performing the PLA reaction, we immunostained the cells with fluorescent secondary antibodies to detect lysosomes labelled by LAMP1 antibody. As a negative control, we omitted the anti-V5 antibody. Coverslips were mounted using the Prolong Gold antifade mounting medium (Thermofisher) instead of the Duolink in situ mounting medium provided with the kit. Images were acquired with a confocal microscope.

## Statistics

Data were analyzed using GraphPad Prism version 9 software. Comparisons of the means of replicate experiments were performed by paired $t$ tests or ANOVA followed by multiple comparisons test following recommendations [66].

## Supporting information

**S1 Fig. Spatacsin is oresent in the ER.** (**A**). Membrane fractions of $Spg11^{+/+}$ and $Spg11^{-/-}$ mouse brains were resuspended in the indicated buffers or detergents and refractionated into the supernatant (S) and membrane pellet (P). Spatacsin was released from membranes only with the detergent deoxycholate (DOC), like the transmembrane protein STIM1. (**B**). MEFs expressing spatacsin constructs with an N-terminal GFP tag or a C-terminal V5-tag. Cells were immunostained with anti-V5 antibody, anti-GFP antibody, and the lysosome marker Lamp1. Scale bar: 5 μm. (**C**) STED images of MEFs expressing V5-tagged spatacsin. Cells were

immunostained with anti-V5, anti-endogenous ER protein REEP5, and anti-LAMP1 antibodies. Scale bar: 1 μm. Arrowheads point spatacsin-V5 colocalized with the ER marker REEP5. Spatacsin V5 occasionnaly colocalized with the lysosome marker Lamp1 and the ER marker REEP5 (arrows). (**D**) Live imaging of the ER marker GFP-Sec61β and lysosome marker Lamp1-mCherry in $Spg11^{+/+}$ and $Spg11^{-/-}$ MEFs. Note that the absence of spatacsin ($Spg11^{-/-}$) did not alter ER morphology. Scale bar: 5 μm. (**E**) Quantification of the average number of lysosomes per square micrometer in $Spg11^{+/+}$ and $Spg11^{-/-}$ MEFs. Superplot: means and SEM, $N$ = 14 cells from 4 independent experiments. Paired $t$ test on the means. (**F**) Quantification of the average lysosomal size per cell in $Spg11^{+/+}$ and $Spg11^{-/-}$ MEFs. Superplot: means and SEM, $N$ = 14 cells from 4 independent experiments. Paired $t$ test on the means. (**G**) Quantification of the proportion of lysosomes that have an area overlapping with the ER > 30% in $Spg11^{+/+}$ and $Spg11^{-/-}$ MEFs when lysosomal staining was flipped by 90˚. Superplot: means and SEM, $N$ = 14 cells from 4 independent experiments. Paired $t$ test on the means. Note that the values are much lower than the ones in Fig 1I, showing that the overlap observed in Fig 1I is not due to random colocalization. The raw data underlying panels E, F, and G can be found in S1 Data file. (PDF)

**S2 Fig. Spatacsin regulates the dynamics of acidic tubular lysosomes.** (**A**). Live imaging of lysosomes stained with various markers in $Spg11^{+/+}$ MEFs. Note that tubular lysosomes (white arrows) were positive for Lamp1, 10 kDa Dextran-Texas Red (TR), Lysotracker green, as well as DQ-BSA green, indicating that they are acidic and catalytically active compartments. Scale bar: 5 μm. (**B**) Quantification of the number of tubular lysosomes in $Spg11^{+/+}$ and $Spg11^{-/-}$ MEFs using the fluorescent markers DQ-BSA, Lysotracker, or Dextran. Superplot: means and SEM, $N$ > 40 cells from 3 independent experiments. Paired $t$ tests on the means. (**C**) Quantification of the proportion of lysosomes with an average speed > 0.3 μm/s according to their shape in wild-type MEFs. Mean and SEM, paired $t$ test. (**D**) Quantification of the average speed of round lysosomes in $Spg11^{+/+}$ and $Spg11^{-/-}$ MEFs. Superplot: means and SEM, $N$ > 87 cells from 4 independent experiments. Paired $t$ test on the means. The raw data underlying panels B, C, and D can be found in S1 Data file. (PDF)

**S3 Fig. Domain of spatacsin encoded by exons 32 to 34 of Spg11 is important for the function of spatacsin.** (**A**) Diagram showing the genomic structure of the mouse $Spg11$ gene (top), the targeting vector (middle), and the targeted locus upon excision of the neomycin resistance cassette and action of the Cre-recombinase (bottom). Numbers indicate exons. The mutations introduced in exon 32 were c.6052C > T (p.Arg2018*) and c.6061C > T (p.Gln2021*). Scheme adapted from [22]. (**B**) Sequencing of RT-PCR product obtained from the brains of homozygous mice that incorporated the targeting vector, showing the splicing of exons 32, 33, and 34. (**C**) Scheme representing the mRNA produced in a wild-type mouse, a mouse that incorporated the targeting vector, or after the action of the Cre recombinase. Note that the intermediate model expressing the floxed allele showed splicing of exons 32 to 34 with conservation of the reading frame between exons 31 and 35. It was thus equivalent to a functional deletion of exons 32 to 34, leading to expression of a protein called Spatacsin$^{Δ32-34}$. (**D**) Western blot showing expression of truncated spatacsin in $Spg11^{Δ32-34/Δ32-34}$ mouse brain. Equal loading was validated by clathrin heavy chain (HC) immunoblotting. (**E**) The time spent on accelerating rotarod was lower in $Spg11^{-/-}$ and $Spg11^{Δ32-34/Δ32-34}$ mice compared to $Spg11^{+/+}$ mice from 4 months of age. Yet, $Spg11^{Δ32-34/Δ32-34}$ mice performance was better than the one of $Spg11^{-/-}$ mice. $N$ = 9 to 15 animals/genotype/age; two-way ANOVA followed by Holm–Sidak post hoc test; $^*P ≤ 0.05$, $^{**}P ≤ 0.01$ and $^{***}P ≤ 0.001$ vs. wild-type ($Spg11^{+/+}$) mice. (**F**) $Spg11^{-/-}$ and $Spg11^{Δ32-34/Δ32-34}$ mice presented a significantly lower performance from the age

of 4 months in the Y-maze test used to evaluate cognitive function. $N = 9$ to 15 animals/genotype/age; two-way ANOVA followed by Holm–Sidak post hoc test; **$P \leq 0.01$ and ***$P \leq 0.001$ vs. wild-type ($Spg11^{+/+}$) mice. (**G**) P62 immunostaining in cortex of 8-month-old $Spg11^{+/+}$, $Spg11^{-/-}$, and $Spg11^{\Delta32-34/\Delta32-34}$ mice. Scale bar: 50 µm. Left: quantification of the number of p62 aggregates per unit area. Note that both $Spg11^{-/-}$ and $Spg11^{\Delta32-34/\Delta32-34}$ cortices presented a high and similar number of p62-positive aggregates. (**H**) Immunostaining of cells expressing the ER marker GFP-Sec61β and V5-tagged spatacsin or V5-tagged spatacsin$^{\Delta32-34}$. Cells were immunostained with anti-V5 antibody and the lysosome marker cathepsin D. Scale bar: 5 µm. The raw data underlying panels E, F, and G can be found in S1 Data file. (PDF)

**S4 Fig. Effect of down-regulation of putative interactors of the domain encoded by exons 32–34 of *Spg11*.** (**A**, **B**) Western blots of wild-type MEFs transfected with control siRNA or independent siRNA that down-regulate spatacsin purchased from Dharmacon (**A**) or ThermoFisher (**B**). Lysate of $Spg11^{-/-}$ MEFs was used as a negative control. Equal loading was validated by clathrin heavy chain (**A**) or α-tubulin (**B**) immunoblotting. (**C**) Graphs showing the mean probability of MEFs transfected with indicated siRNA to be considered as a knockout by the trained neural network, based on the analysis of the lysosomal staining with Dextran-Texas Red. Two independent siRNAs were tested for each target gene and 3 for *Spg11*. Dashed line corresponds to the minimal effect obtained with an siRNA down-regulating *Spg11*. (**D**) Graph showing the mean tubulation scores when candidates interactors of spatacsin were down-regulated by siRNA. Dashed line corresponds to the minimal effect obtained with an siRNA down-regulating *Spg11*. The raw data underlying panels C and D can be found in S1 Data file. (PDF)

**S5 Fig. Lysosomal localization of GFP-mCherry-AP5Z1.** (**A**) Quantification of AP5M1 levels by western blot (Fig 4A): means and SEM, $N = 3$ independent experiments. Kruskal–Wallis test. (**B**) Western blot with anti-AP5Z1 antibody and anti-HA antibody upon transfection of $Spg11^{+/+}$ and $Spg11^{-/-}$ MEFs with vector overexpressing AP5Z1-His and spastizin-HA. Right: quantification of relative levels of AP5Z1-His and Spastizin-HA. Means and SEM, $N = 4$ independent experiments. Mann–Withney test. (**C**) Live images of $Spg11^{+/+}$ and $Spg11^{-/-}$ MEFs expressing GFP-mCherry-AP5Z1 treated with bafilomycin 100 nM for 16 hours. Note that GFP and mCherry signals perfectly colocalize. Scale bar: 10 µm. The raw data underlying panels A and B can be found in S1 Data file. (PDF)

**S6 Fig. AP5Z1 and spastizin compete for interaction with spatacsin.** (**A**) Negative control for the proximity ligation assay (PLA) used to show the interaction between V5-tagged spatacsin and HA-tagged spastizin (Fig 5C). The PLA signal (magenta) is almost absent. Scale bar 10 µm. (**B**) Co-immunoprecipitation of spastizin and AP5Z1 with spatacsin-GFP upon overexpression of AP5Z1. Input represents 5% of lysate added to the immunoprecipitation assay. Inputs and immunoprecipitates were loaded on 2 separate gels processed simultaneously. Note that spatacsin and spastizin looked slightly different in input and co-immunoprecipitation, likely due to the high amount of both proteins in immunoprecipitates. Right: quantification of the amount of AP5Z1 in the input, as well as the relative amount of AP5Z1 or spastizin co-immunoprecipitated with Spatacsin-GFP. Means and SEM, $N = 4$ independent experiments. Mann–Whitney test. Note that the interaction of spatacsin-GFP with spastizin decreases when AP5Z1 is overexpressed. (**C**) Quantification of the proportion of spastizin-GFP colocalized with Lamp1-mCherry in wild-type MEFs overexpressing AP5Z1. Superplot:

means and SEM, $N > 53$ cells from 3 independent experiments. Paired $t$ test on the means. The raw data underlying panels B and C can be found in S1 Data file.
(PDF)

**S7 Fig. KIF13A and p150^Glued contribute to lysosome trafficking.** (**A**) Western blot of wild-type MEFs transfected with a control siRNA or siRNA down-regulating AP5Z1. (**B**) Quantification of the number of tubular lysosomes in wild-type MEFs transfected with a vector overexpressing GFP-AP5Z1. Superplot: means and SEM, $N > 78$ cells from 5 independent experiments. Paired $t$ test on the means. (**C**) Quantification of the proportion of tubular lysosomes moving >1.2 µm over 1 minute in wild-type MEFs transfected with a vector overexpressing GFP-AP5Z1. Superplot: means and SEM, $N = 30$ cells from 3 independent experiments. Paired $t$ test on the means. (**D**) Western blot of wild-type MEFs transfected with a control siRNA or siRNA down-regulating Spg15. (**E**) Images of wild-type MEFs expressing spastizin-GFP and Lamp1-mCherry treated with 100 nM wortmannin for 1 hour. Note the loss of colocalization of spastizin-GFP and Lamp1-mCherry upon wortmannin treatment. Scale bar: 5 µm. (**F**) Quantification of the number of tubular lysosomes in wild-type MEFs treated with wortmannin. Superplot: means and SEM, $N > 32$ cells from 3 different independent experiments. Paired $t$ test on the means. (**G**) Western blots showing co-immunoprecipitation of spastizin-HA, but not spastizin-ΔCT (lacking aa 2,120–2,539) with KIF13A-ST-YFP. Input represents 5% of lysate added to the immunoprecipitation assay. (**H**) Quantification of the number of tubular lysosomes in wild-type MEFs transfected with siRNA targeting mouse *Spg15* (si Spg15) and either GFP, human spastizin-GFP, or human spastizinΔCT-GFP. Superplot: means and SEM, $N > 70$ cells from 5 independent experiments. RM one-way ANOVA on the means, Holm–Sidak's multiple comparisons test. (**I**) Quantification of the proportion of tubular lysosomes moving >1.2 µm over 1 minute in wild-type MEFs with siRNA targeting mouse *Spg15* (si Spg15) and either GFP, human spastizin-GFP, or human spastizinΔCT-GFP. Superplot: means and SEM, $N >45$ cells from 4 independent experiments. RM one-way ANOVA on the means, Holm–Sidak's multiple comparisons test. The raw data underlying panels B, C, F, H, and I can be found in S1 Data file.
(PDF)

**S8 Fig. AP5Z1 contributes to lysosome directionality.** (**A**) Live image of primary cortical neuron transfected with Lamp1-GFP and mCherry-TRIM46 that labels the axon initial segment, allowing us to identify axons (white arrows). Scale bar: 10 µm. (**B**) Quantification of the proportion of lysosomes that are moving retrogradely along the axon of $Spg11^{+/+}$ and $Spg11^{-/-}$ primary mouse neurons. Superplot: means and SEM, $N > 79$ cells from 5 independent experiments. Paired $t$ test on the means. (**C**) Quantification of the proportion of lysosomes that are moving anterogradely along the axon of $Spg11^{+/+}$ primary mouse neurons expressing either GFP or GFP-AP5Z1. Superplot: means and SEM, $N > 32$ cells from 3 independent experiments. Paired $t$ test on the means. (**D**) Quantification of the proportion of lysosomes that are moving retrogradely along the axon of $Spg11^{+/+}$ primary mouse neurons expressing either GFP or GFP-AP5Z1. Superplot: means and SEM, $N > 32$ cells from 3 independent experiments. Paired $t$ test on the means. The raw data underlying panels B, C, and D can be found in S1 Data file.
(PDF)

**S1 Video. Live imaging of wild-type MEF expressing Lamp1-mCherry and Sec61β-GFP.** Note the tubular lysosome movement along ER tubular network. One frame was acquired every 750 ms. The video is sped up 4 times. Scale bar: 3 µm.
(AVI)

**S2 Video. Live imaging of _Spg11_$^{+/+}$ MEF expressing Lamp1-mCherry in green.** Detected tubular endolysosomes are highlighted in magenta. One frame was acquired every 750 ms. The video is sped up 4.5 times. Scale bar: 10 μm.
(MP4)

**S3 Video. Live imaging of _Spg11_$^{-/-}$ MEF expressing Lamp1-mCherry in green.** Detected tubular endolysosomes are highlighted in magenta. One frame was acquired every 750 ms. The video is sped up 4.5 times. Scale bar: 10 μm.
(MP4)

**S1 Table. List of preys identified by a yeast two-hybrid screen using C-terminal domain of human SPG11 (aa 1,943–2,443).** Confidence in interaction: A, very high confidence; B, high confidence; C, good confidence; D, moderate confidence.
(DOCX)

**S2 Table. List of preys identified by a yeast two-hybrid screen using C-terminal domain of human SPG11 lacking domain encoded by exons 32–34.** Confidence in interaction: A, very high confidence; B, high confidence; C, good confidence; D, moderate confidence.
(DOCX)

**S3 Table. Unbiased analysis of the effect of siRNA down-regulating genes encoding putative binding partners of domain of spatacsin encoded by exons 32–34 of _SPG11_.** The scores represent the probablility of phenocopying _Spg11_$^{-/-}$ MEFs (see methods). Bold indicates genes that are at least as efficient as 3 independent Spg11 siRNA (SPG11#1: 0.36; SPG11#2: 0.21; SPG11#3: 0.26).
(DOCX)

**S4 Table. Analysis of the proportion of tubular lysosomes in control MEFs transfected with siRNA down-regulating genes encoding putative binding partners of domain of spatacsin encoded by exons 32–34 of SPG11.** The scores represent the normalized number of tubules (score = 1 for control MEFS, score = 0 for _Spg11_$^{-/-}$ MEFS). Bold indicates genes that are at least as efficient as 3 independent Spg11 siRNA (SPG11#1: 0.55; SPG11#2: 0.43; SPG11#3: 0.50) to decrease the proportion of tubular lysosomes.
(DOCX)

**S5 Table. List of siRNA used for the screen (Fig 3) in the study.**
(XLS)

**S1 Data. Excel file containing the underlying numerical data for Figs 1E, 1G, 1H, 1J, 2B, 2C, 2F, 2G, 2H, 2I, 3B, 3C, 4A, 4C, 4E, 5B, 5C, 6B, 6C, 6D, 6E, 7A, 7B, 7C, 7E, 7F, 7H, 7I, 8B, 8C, 8E, 8F, 8G, 8H, 8I, S1E, S1F, S1G, S2B, S2C, S2D, S3E, S3F, S3G, S4C, S4D, S5A, S5B, S6B, S6C, S7B, S7C, S7F, S7H, S7I, S8B, S8C and S8D.**
(XLSX)

**S1 Raw Images. Uncropped western blots related to Figs 1A, 1B, 4A, 4B, 4C, 5A, 5B, 5C, 6D, 7D, 7G, S1A, S3D, S4A, S4B, S5B, S6B, S7A, S7D and S7G.**
(PDF)

## Acknowledgments

We thank the Phenoparc, Celis, iGenSeq, and ICM.quant core facilities of the Paris Brain Institute and Noemi Asfogo for their contributions. The NeurImag Imaging Facility team (part of

IPNP, Inserm U1266 and Université Paris Cité) thanks the Leducq Foundation for supporting the acquisition of the IPNP Leica SP8 Confocal/STED 3DX microscope.

## Author Contributions

**Conceptualization:** Alexandre Pierga, Julien Branchu, Maxime Boutry, Frédéric Darios.

**Formal analysis:** Alexandre Pierga, Julien Branchu, Lydia Danglot, Maxime Boutry, Frédéric Darios.

**Funding acquisition:** Frédéric Darios.

**Investigation:** Margaux Cauhapé, Julien Branchu, Maxime Boutry.

**Methodology:** Alexandre Pierga, Raphaël Matusiak.

**Project administration:** Frédéric Darios.

**Resources:** Frédéric Darios.

**Software:** Raphaël Matusiak.

**Writing – original draft:** Alexandre Pierga, Raphaël Matusiak, Frédéric Darios.

**Writing – review & editing:** Alexandre Pierga, Frédéric Darios.

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
