## [Editor Report · Decision Letter 0]

28 Dec 2022

Dear Dr Darios, 

Thank you for submitting via Review Commons your manuscript entitled "Spatacsin regulates directionality of lysosome trafficking" for consideration as a Research Article by PLOS Biology.

Your manuscript has now been evaluated by the PLOS Biology editorial staff as well as by an academic editor with relevant expertise and I am writing to let you know that we would like to invite you to submit a revision.

However, before we can do that, we need you to complete your submission by providing the metadata that is required for further consideration. To this end, please login to Editorial Manager where you will find the paper in the 'Submissions Needing Revisions' folder on your homepage. Please click 'Revise Submission' from the Action Links and complete all additional questions in the submission questionnaire.

Once your full submission is complete, your paper will undergo a series of checks in preparation for inviting the revision. After your manuscript has passed the checks I will send you the decision. To provide the metadata for your submission, please Login to Editorial Manager (https://www.editorialmanager.com/pbiology) within two working days, i.e. by Dec 30 2022 11:59PM.

Kind regards,

Ines

--

Ines Alvarez-Garcia, PhD

Senior Editor

PLOS Biology

---

## [Editor Report · Decision Letter 1]

4 Jan 2023

Dear Dr Darios,

Thank you for completing the submission of your Review Commons manuscript entitled "Spatacsin regulates directionality of lysosome trafficking" to PLOS Biology. It has now been evaluated along with the reviews by the PLOS Biology editors and an Academic Editor with relevant expertise.

In light of this advice, we would like to invite you to revise the work to address the remaining reviewers' reports along the lines you indicate in the rebuttal. Your revised manuscript is likely to be sent for further evaluation by all or a subset of the reviewers.

**IMPORTANT - SUBMITTING YOUR REVISION**

3. Resubmission Checklist

a) *PLOS Data Policy*

b) *Published Peer Review*

d) *Blurb*

Please also provide a blurb which (if accepted) will be included in our weekly and monthly Electronic Table of Contents, sent out to readers of PLOS Biology, and may be used to promote your article in social media. The blurb should be about 30-40 words long and is subject to editorial changes. It should, without exaggeration, entice people to read your manuscript. It should not be redundant with the title and should not contain acronyms or abbreviations. For examples, view our author guidelines: https://journals.plos.org/plosbiology/s/revising-your-manuscript#loc-blurb

Sincerely,

Ines

--

Ines Alvarez-Garcia, PhD

Senior Editor

PLOS Biology

---

## [Decision Letter · Decision Letter 2]

13 Jul 2023

Dear Dr Darios,

Thank you for your patience while we considered your revised manuscript entitled "Spatacsin regulates directionality of lysosome trafficking" for publication as a Research Article at PLOS Biology. Your revised study has been evaluated by the PLOS Biology editors, the Academic Editor and three of the original reviewers.

The reviews are attached below. As you will see, while the reviewers appreciate the improvements you have done in the revision, they still raise several concerns that have not been addressed. Reviewer 3 thinks that the connection between spatacsin and UBR4 and the role in AP5Z1 degradation is not convincing and suggests that you can directly test if downregulation of UBR4 leads to increase stability of spatacsin in a CHX chase experiment that could be performed on the endogenous proteins. In addition, this reviewer would like you to repeat the IP showing the interaction between spatacsin and UBR4. Reviewer 4 feels that many results are overinterpreted and should be toned down, and that the trafficking effects observed could add to morphologic effects or reflect that the proteins are downstream.

In light of the reviews and consultation with the Academic Editor, we have decided to give you one last chance to revise the work to thoroughly address the remaining points raised by the reviewers. Given the extent of revision needed, we cannot make a decision about publication until we have seen the revised manuscript and your response to the reviewers' comments. Your revised manuscript is likely to be sent for further evaluation by all or a subset of the reviewers.

**IMPORTANT - SUBMITTING YOUR REVISION**

3. Resubmission Checklist

a) *PLOS Data Policy*

b) *Published Peer Review*

c) *Blurb*

Please also provide a blurb which (if accepted) will be included in our weekly and monthly Electronic Table of Contents, sent out to readers of PLOS Biology, and may be used to promote your article in social media. The blurb should be about 30-40 words long and is subject to editorial changes. It should, without exaggeration, entice people to read your manuscript. It should not be redundant with the title and should not contain acronyms or abbreviations. For examples, view our author guidelines: https://journals.plos.org/plosbiology/s/revising-your-manuscript#loc-blurb

Sincerely,

Ines

--

Ines Alvarez-Garcia, PhD

Senior Editor

PLOS Biology

Reviewers' comments

Rev. 1:

The authors have dealt with my comments in a very comprehensive fashion and this is now a very convincing story that I feel will be a significant advance for the field. I have a few very minor points to be considered:

I can see a faint spatacsin band in figure 1B lysosomal fraction, so the comment about spatacsin not being on degredative lysosomes should be softened - especially as the tubular lysosomes whose movement is controlled by spatacsin seem to be catalytically active (as shown in figure 2).

Figure 1D - please add arrow and arrowheads to all images in the panel (not just two), for better orientation.

In suppl fig 3G the marker labelled should be stated on the micrograph panels.

For the description of the results in figure 4C it would be more correct to say "partially prevented" the degradation of AP5Z1, as some downregulation still seems to happen.

Figure 7H and I should be swapped around so that they can be called from the text in the correct order.

These comments are so trivial that I do not feel that I would need to see another revised version. I would like to congratulate the authors on their interesting work, which has been conducted and presented in a thorough and comprehensive way.

Rev. 3:

The authors have put a lot of efforts to answer to the previous concerns. They have also down-played some of the previous conclusions. The results on the lysosomal phenotypes are convincing. In general, I think that the manuscript has improved. However, I still think that the connection between spatacsin and UBR4 and the role in the degradation of AP5Z1 is weak and not convincing (see point 1). If spatacsin has a role in lysosomal tubulation, this can affect AP5Z1 stability independently from a molecular interaction with UBR4.

1) I am still not convinced about the role of UBR4, as proposed by the authors. UBR4 as the other putative interactors with a role in lysosomal tubulation have been identified by yeast-two hybrid, a technique that forces the cytosolic spatacsin protein to the nucleus. It is possible that this leads to increased degradation, owing to the unphysiological localization, and enriches "interactors" linked to this pathway (there is also a proteasomal subunit). In fact, in the screen, UBR4 appears with a moderate score (D). Obviously, these "interactors" would lead to a lysosomal phenotype when depleted. The results in Figure 4 are still unconvincing. Despite the quantification, I cannot see any difference in control condition in Fig. 4C upon UBR4 downregulation. The authors can directly test if downregulation of UBR4 leads to increase stability of spatacsin in a CHX chase experiment This experiment can even be performed on the endogenous proteins. Is AP5Z1 more ubiquitinated when spatacsin is overexpressed? The IP showing the interaction of spatacsin and UBR4 is also problematic, since the input is not loaded on the same gel, so we cannot judge how efficient is this IP and the control still shows the interaction.

2) Figure 5D. Same as above. In addition, less spatacsin-GFP is immunoprecipitated in the condition of downregulation of UBR4, so it is expected that also less spastizin is pulled-down. Also, why are the bands in the input and IP so different for spatacsin and spastizin?

3) The superplots show now quite clearly that some effects are barely significant (only paired t-test). Of course, this does not mean that these data are not real, but the authors should be aware that there is a certain risk of overinterpretation. The data may become more or less solid with additional experiments. Still, this is a honest way of showing the data. I would recommend to directly show the real p value on the plot.

Rev. 4:

The authors have done an earnest job of reformatting the data as suggested by other reviewers, plotting individual experiment averages and re-analyzing the data which have added to the strength of the findings. The quality of some blots has also improved.

However, the revised manuscript still falls short of convincing mechanistic insight and new data were not sufficiently provided to address these concerns across the reviewers. Clearly, the loss of spatascin results in less numerous tubular lysosomes - this is seen nicely throughout the paper. But the mechanisms behind spatascin doing this are less well defined. Many details are over-interpreted. Spatascin is found in ER, but is not enriched as the authors claimed - there is lots of spatascin elsewhere in the cell (imaging) and the fractionation of spatascin does not follow STIM1 enrichment, for example. It is localized to ER but the incongruent data for this model is a bit overlooked. UBR4 is a cytosolic E3 ligase, but they authors argue with limited data that it mediates degradation of AP5Z1 (sensitive to bafilomycin, again not well explained) where compensatory/transcriptional effects are not considered (mRNA) and the degradation evidence is incomplete. Since morphology alone (round vs tubular) seems to affect trafficking, it is not entirely clear that the trafficking effects are in addition to morphologic effects (additive) or merely downstream. Likewise, the motor protein interaction and trafficking data are not sufficient to support the robust conclusions.

Rather than focus on a subset of protein-protein interactions and precise mechanistic principles, the manuscript seeks to conclude, in large sweeping statements, the function and inter-molecular interactions of about a half a dozen proteins and falls slightly short throughout. The mechanisms of affecting lysosome morphology, protein recognition and degradation, and motor-protein association are each limited.

---

## [Editor Report · Decision Letter 3]

11 Sep 2023

Dear Dr Darios,

Thank you for your patience while we considered your revised manuscript entitled "Spatacsin regulates directionality of lysosome trafficking" for publication as a Research Article at PLOS Biology. This revised version of your manuscript has been evaluated by the PLOS Biology editors and by the Academic Editor.

Based on our Academic Editor's assessment of your revision, we are likely to accept this manuscript for publication, provided you satisfactorily address the data and other policy-related requests stated below.

In addition, we would like you to consider a suggestion to improve the title:

"Spatacsin regulates directionality of lysosome trafficking by promoting the degradation of its partner AP5Z1"

We expect to receive your revised manuscript within two weeks. 

*Published Peer Review History*

*Press*

Sincerely,

Ines

--

Ines Alvarez-Garcia, PhD

Senior Editor

PLOS Biology

DATA POLICY:

Many thanks for sending all the data underlying the graphs shown in the figures. Please also indicate where the data can be found in each corresponding figure legend (both main and supplementary figures). The data file uploaded in the editorial system should be named S1_Data.xlsx (using an underscore) and in the figure legends and the manuscript should be described as S1 Data. For example, you can add at the end of the figure legend: "The data underlying the graphs shown in the figure can be found in S1 Data."

**Please also make sure that the data you have deposited in the IntAct database (IM-29701) is publicly available at this stage.

SPECIES INDICATED IN THE ABSTRACT

- Please note that per journal policy, the model system/species studied should be clearly stated in the abstract of your manuscript. Please add this to the title or abstract.

---

## [Editor Report · Decision Letter 4]

15 Sep 2023

Dear Dr Darios,

Thank you for the submission of your revised Research Article entitled "Spatacsin regulates directionality of lysosome trafficking by promoting the degradation of its partner AP5Z1" for publication in PLOS Biology. On behalf of my colleagues and the Academic Editor, Maya Schuldiner, I am delighted to let you know that we can in principle accept your manuscript for publication, provided you address any remaining formatting and reporting issues. These will be detailed in an email you should receive within 2-3 business days from our colleagues in the journal operations team; no action is required from you until then. Please note that we will not be able to formally accept your manuscript and schedule it for publication until you have completed any requested changes.

PRESS

Sincerely, 

Ines

--

Ines Alvarez-Garcia, PhD

Senior Editor

PLOS Biology
